# The genetic landscape of a physical interaction

**Guillaume Diss[1,2], Ben Lehner[1,2,3]\***

[1]Systems Biology Program, Centre for Genomic Regulation, The Barcelona Institute for Science and Technology, Barcelona, Spain; [2]Universitat Pompeu Fabra, Barcelona, Spain; [3]Institució Catalana de Recerca i Estudis Avançats, Barcelona, Spain

**Abstract** A key question in human genetics and evolutionary biology is how mutations in different genes combine to alter phenotypes. Efforts to systematically map genetic interactions have mostly made use of gene deletions. However, most genetic variation consists of point mutations of diverse and difficult to predict effects. Here, by developing a new sequencing-based protein interaction assay – *deepPCA* – we quantified the effects of >120,000 pairs of point mutations on the formation of the AP-1 transcription factor complex between the products of the FOS and JUN proto-oncogenes. Genetic interactions are abundant both in *cis* (within one protein) and *trans* (between the two molecules) and consist of two classes – interactions driven by thermodynamics that can be predicted using a three-parameter global model, and structural interactions between proximally located residues. These results reveal how physical interactions generate quantitatively predictable genetic interactions.

DOI: https://doi.org/10.7554/eLife.32472.001

## Introduction

Mutations often have outcomes that change depending upon additional genetic variation carried by an individual, making their effects difficult to predict (*Lehner, 2011*). The unexpected outcomes obtained when two or more mutations are combined are referred to as genetic interactions or epistasis (*Phillips, 2008*).

One approach that has been taken to better understand how mutations interact to alter phenotypes has been to systematically combine together gene deletions or representative hypomorphic alleles (*Baryshnikova et al., 2013*). In budding yeast, this has been undertaken on a genomic scale, with the resulting network of interactions referred to as the 'genetic landscape' of a cell (*Costanzo et al., 2010*, *2016*; *Tong et al., 2004*).

However, gene deletions are rare in nature – most genetic variation consists of point mutations not deletions or null alleles. Point mutations can have very diverse and difficult to predict effects (*Shendure and Akey, 2015*). These range from no consequence, through partial loss-of-function, to very strong effects or the creation of new functions. To date, however, there has been no systematic effort to map how point mutations in two genes combine together to alter biological functions.

Protein-protein interactions (PPIs) represent the backbone of a cell's functional organization. Mutations affecting PPIs lead to disease, to functional innovations, and hence are subject to selection (*Diss et al., 2013*). It has long been appreciated that pairs of mutations in two physically interacting proteins can have non-additive outcomes (*Horovitz, 1996*; *Lehner, 2011*). However, to date, the effects of mutations on PPIs have only been quantified for deep mutant libraries of one protein in combination with a small number of targeted mutants in a physical interaction partner (*Aakre et al., 2015*; *Araya et al., 2012*; *Raman et al., 2016*). A thorough understanding of the

**\*For correspondence:**
lehner.ben@gmail.com

**Competing interests:** The authors declare that no competing interests exist.

**eLife digest** Proteins, the molecular workhorses of the cell, are made of small units called amino acids attached together like the links of a chain. Each protein is composed of a unique combination of amino acids, which is determined by a specific sequence of DNA called a gene. A change in a gene – a mutation – can create a variation in the protein it codes for, for instance by swapping a type of amino acid for another. Different mutations in the same gene can alter a protein in different ways. Some of these changes are harmless, but other can hinder how the protein performs its role. For example, a small change in the structure of a protein could affect how it will bind to other molecules.

It is possible for people to have identical mutations in the same genes, but experience different consequences. For instance, two persons could carry the same disease-inducing mutation, but one has a severe version of the condition and the other only mild symptoms. One reason is that changes in other genes cancel out or enhance the effect of a mutation. This phenomenon is known as a genetic interaction and it remains poorly understood, especially at the molecular level.

Here, Diss and Lehner developed a method, called *deepPCA*, to study the consequences of mutations in proteins in the laboratory. The experiments focused on two human genes which code for two proteins that normally attach to each other. Two mutations were artificially created, either one in each gene, or two in one of them. Diss and Lehner then examined how strongly the two mutated proteins could still attach to each other. By repeating this process with over 120,000 different pairs of mutations, it became possible to study how one mutation can have different effects depending on the presence of other mutations in the same protein or in the binding partner.

Overall, Diss and Lehner found that genetic interactions are the result of two mechanisms. In the first one, the two mutations together cause specific structural changes that modify how proteins bind to each other. In the second one, the changes solely depend on the magnitude of the initial, thermodynamic effects of individual mutations, but not on their specific physical and chemical properties. To predict the consequences of this second type of genetic interactions, knowing the identity or the exact effects of the two mutations is not necessary.

Understanding and predicting genetic interactions is important to develop personalized medicine, where treatments are tailored based on the genetic make up of an individual. This knowledge will also help to study how genes have evolved together.

DOI: https://doi.org/10.7554/eLife.32472.002

patterns of mutation outcome between interacting proteins requires a non-biased, systematic mutagenesis of both interacting proteins.

Here, we present a high-throughput technique based on the protein fragment complementation assay (*Tarassov et al., 2008*; *Diss et al., 2017*) (PCA) called *deepPCA* that quantifies how mutations of diverse individual effect combine to alter protein interactions. We used the assay to systematically and comprehensively determine the effects of combinations of mutations in the proto-oncogenes *FOS* and *JUN* on the formation of the AP-1 transcription factor complex (*Shaulian and Karin, 2002*). Fos and Jun interact through their leucine zipper domains that consist of five heptad repeats (*Figure 1A*); this interaction has been previously extensively investigated (*Mason et al., 2006*; *Ransone et al., 1989*). We first quantified the consequences of combining thousands of pairs of mutations in *trans* between the two proteins. We then compared these results to the effects of thousands of pairs of mutations in *cis* within one of the proteins (Fos).

The resulting dataset presents a global view of how hundreds of mutations of diverse individual effects in different genes combine to alter a biological function through two major mechanisms related to the thermodynamics of a PPI and the structural interactions between proximal residues.

## Results

### Quantifying thousands of protein interactions in parallel using *deepPCA*

To quantify how mutations of diverse individual effect combine to alter protein interactions, we developed *deepPCA*, a protein-protein interaction assay that uses PCA and deep sequencing to

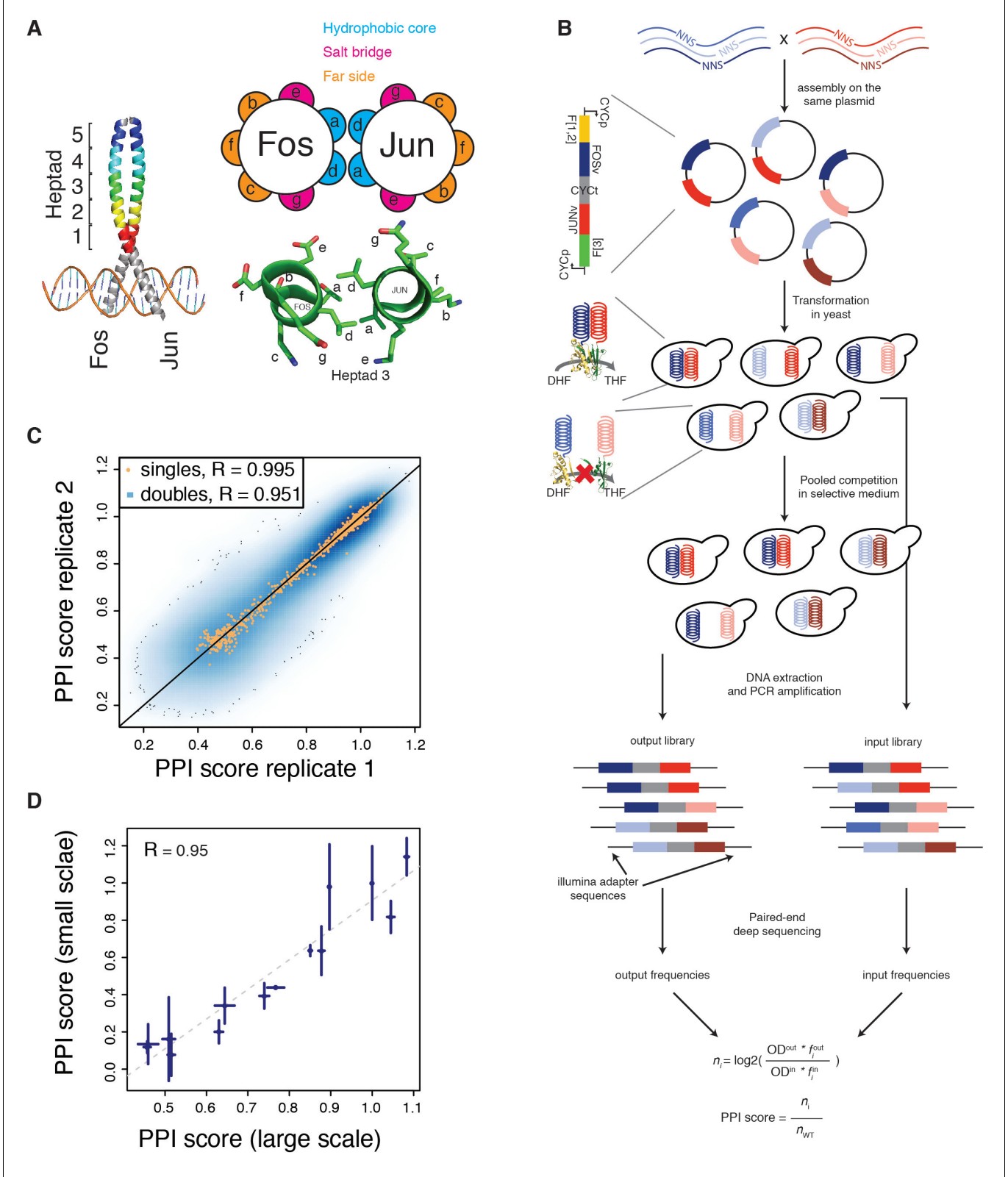

**Figure 1.** *deepPCA.* (**A**) Leucine zipper domains (colored) and heptad positions of human Fos and Jun. (**B**) Overview of the assay. Single amino acid variants of Fos and Jun were constructed by overlap-extension PCR using NNS primers and cloned in a head-tail orientation. In PCA, interacting proteins expressed in yeast lead to complementation between their fused DHFR fragments, which is resistant to methotrexate and produces tetrahydrofolate (THF) from dihydrofolate (DHF) to promote growth. Paired-end deep sequencing then allows the frequencies of each variant in the

*Figure 1 continued on next page*

*Figure 1 continued*

input and output populations to be measured and a PPI score that represents the number of generation of each variant relative to the wild-type interaction to be computed. (**C**) Scatter plot of PPI scores between biological replicates 1 and 2. (**D**) Confirmation of single mutants by individual PCA growth assays. Single mutants were reconstructed, sequence confirmed and their PPI scores were derived from their growth curves measured in a plate reader (see Materials and methods). Error bars represent 95% confidence intervals.

DOI: https://doi.org/10.7554/eLife.32472.003

The following figure supplement is available for figure 1:

**Figure supplement 1.** Quality control of the PPI scores measured by *deepPCA*.

DOI: https://doi.org/10.7554/eLife.32472.004

quantify thousands of protein-protein interactions in parallel in a single assay (*Figure 1B*, see Materials and methods). *deepPCA* uses deep sequencing to quantify the effects on a PPI of thousands of combinations of point mutations within one or both physically interacting proteins. The method is inspired by deep mutation scanning experiments on individual proteins (*Fowler et al., 2010*; *Fowler and Fields, 2014*) and uses physical linkage on a plasmid to read out the frequency of each pair of mutations after a competitive selection for growth dependent on the physical interaction between two proteins (*Figure 1B*, see Materials and methods). Briefly, the two proteins of interest are fused to complementary halves of a methotrexate-resistant variant of murine dihydrofolate reductase and expressed in yeast. If the two proteins interact, the two fragments complement each other and reform an active enzyme, allowing growth in the presence of methotrexate. PCA is highly quantitative because the growth rate is correlated to the abundance of the complementation complex (*Freschi et al., 2013*; *Levy et al., 2014*; *Schlecht et al., 2012*) so cells expressing strongly interacting variants of the two proteins will hence grow faster and be enriched in the population while cells expressing weakly interacting variants and variants that don't interact will be depleted. These changes in frequency between the pre- and post-selection populations (input and output, respectively) are then quantified by paired-end deep sequencing. The final PPI score quantifies the strength of interaction relative to the wild-type protein (*Figure 1B*).

We used *deepPCA* to quantify the effects of systematically mutating the leucine zipper domains of *FOS* and *JUN*. We obtained reliable (input reads > 10 and output reads > 0; *Figure 1—figure supplement 1A,B*; see Materials and methods) measurements for 607 and 608 of the 608 (32 positions x 19 substitutions) possible single amino acid (aa) changes within the targeted regions of Fos and Jun, respectively. PPI scores measured by *deepPCA* are highly reproducible between biological replicates (mean Pearson correlation R = 0.95 between the three pairs of replicates, n = 108,840 mutation combinations, *Figure 1C*, *Figure 1—figure supplement 1C* and *Supplementary file 1*) and also with mutation effects tested individually (R = 0.95 for 14 variants chosen randomly, *Figure 1D*).

The PPI scores for single amino acid changes in both proteins show a bimodal distribution (*Figure 2—figure supplement 1A*), with ~20% and 15% of substitutions severely detrimental for the interaction and significantly different from the wild-type (PPI score $\leq$ 0.64, FDR < 0.05, one sample t-test against a mean of 1; *Figure 2—figure supplement 1B*). However, the individual substitutions altered the interaction across the entire dynamic range, with 25 and 10 aa changes in each protein strengthening the interaction (PPI score > 1.04, FDR < 0.05, average SEM of these 35 variants = 0.0054; *Figure 2—figure supplement 1C*).

## Determinants of single mutant outcome

Mutations in the hydrophobic core of the interaction interface (heptad positions *a* and *d*) are most detrimental, followed by mutations at salt-bridge positions (positions *e* and *g*, *Figure 2A–B*). Mutations in the hydrophilic far side of the zipper (positions *b, c* and *f*) were generally of small effect (*Figure 2A–C*). Changes in the physico-chemical properties of the amino acids (hydrophobicity, charge, α-helical stability etc, see *Supplementary file 2*) provide good prediction of the mutation effects (percentage of variance explained from 35% to 98% across Fos and Jun positions), with properties related to α-helical stability most informative for predicting single mutation outcomes (*Figure 2—figure supplement 1D* and *Supplementary file 2*).

Identical substitutions in the same positions in Fos and Jun often had similar effects. For instance, mutations in one protein that disrupted the interaction (PPI scores < 0.64) were also very likely to

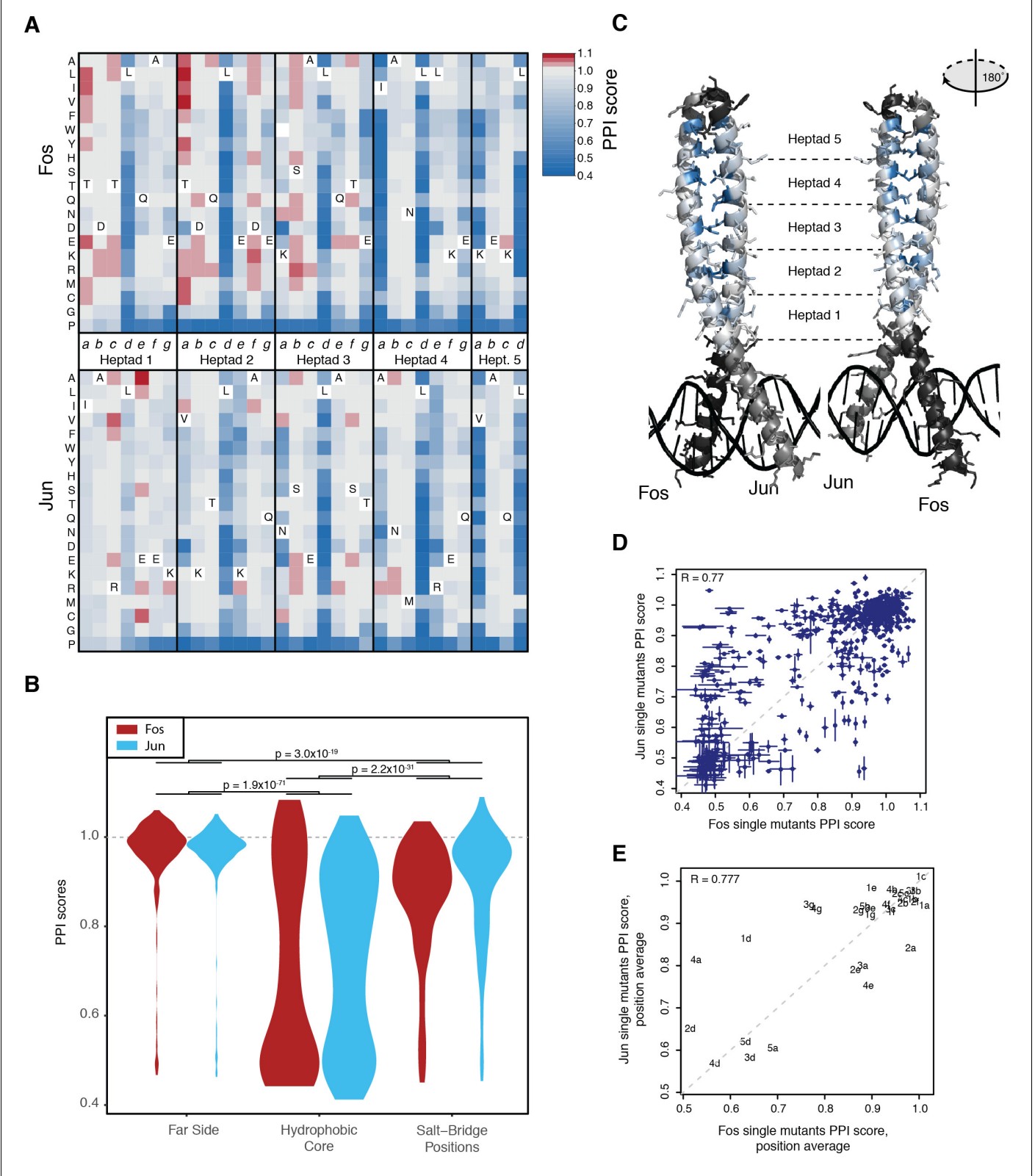

**Figure 2.** Effects of single mutants. (**A**) Heatmap of single mutant PPI scores averaged between the three replicates. Letters inside the heatmap represent the wild-type amino acid. White represents missing data. (**B**) Distribution of PPI scores per position types. p-Values from Welch t-test. (**C**) Average PPI score per position overlaid on the crystal structure (pdb: 1fos). Black and gray represent positions not mutated in Fos and Jun, respectively. (**D**) Scatter plot between PPI scores of corresponding single mutations at the same positions in Fos and Jun. Error bars represent 95%

*Figure 2 continued on next page*

*Figure 2 continued*

confidence intervals. (E) Scatter plot between average PPI score per corresponding positions in Fos and Jun. The number represents the heptad and the letter the position inside the heptad.

DOI: https://doi.org/10.7554/eLife.32472.005

The following figure supplement is available for figure 2:

**Figure supplement 1.** Single mutant effects.

DOI: https://doi.org/10.7554/eLife.32472.006

disrupt the interaction when made in the other protein (odds ratio = 31.6, p<$2.2\times10^{-16}$, Fisher's exact test; *Figure 2D*). Near neutral or strengthening mutations (PPI scores > 0.96) in one protein were also more likely to have a similar effect in the other one (odds ratio = 7.3, p < $2.2\times10^{-16}$, Fisher's exact test). However, a substantial number of substitutions had effects that differed between the two proteins (n = 381 out of 581, FDR < 0.05, paired t-test between the three replicate measurements in Fos and Jun), underlining the importance the structural context in which they occur. These mutations are enriched in intermediate effects in one or both proteins (odds ratio = 8.1, p < $2.2\times10^{-16}$, Fisher's exact test). The average PPI score per position was also generally conserved between the two proteins, but revealed positions asymmetrically involved in the interaction such as the salt bridge positions (*Figure 2E*).

### *trans* genetic interactions between mutations in Fos and Jun

Considering pairs of substitutions in the two proteins, we obtained data for 107,625 of the 369,664 possible double mutants (input read count above 10 and output read count above 0, *Supplementary file 1*, see Materials and methods). The double mutant PPI scores also show a bimodal distribution, but with proportionally more severely detrimental (~26%) and fewer near-neutral outcomes (~21%) than for the single mutants (*Figure 3A*).

The outcome of the double mutations was well predicted by multiplying the PPI scores of the constitutive single mutants (percentage of variance explained of 85–86% in all three replicates, *Figure 3B*), that is, by assuming no genetic interaction between mutations. We calculated a genetic interaction score for each double mutant as the difference between the observed and predicted PPI scores (*Supplementary file 1*). Negative and positive genetic interactions (16,394 and 11,653 cases, respectively, at a 20% FDR, one-sample t-test) thus represent double mutants with lower or higher interaction strength than expected, respectively. The genetic interaction scores are well correlated between replicates with a distribution centered on zero and long tails of positive and negative scores (*Figure 3—figure supplement 1*). Thus, as observed in other systems (*Araya et al., 2012*; *Olson et al., 2014*), genetic interactions make an important contribution to the outcome of double mutations.

### Global dependencies in the genetic interaction landscape

The genetic interaction scores are, however, strongly dependent on the single mutant PPI scores (*Figure 3C*). Combining two mutants that both moderately reduce PPI strength is likely to result in a negative genetic interaction (*Figure 3C*). Positive genetic interactions are, however, generally detected between two mutations that greatly weaken the interaction and also often when combining strength-increasing and strength-decreasing mutations (*Figure 3C*).

### A thermodynamic model accurately predicts double mutant outcome

To account for these trends, we considered the thermodynamics of a PPI, relating the concentration of the bound and total subunits to the free energy of a dimeric complex (*equation 9* in the Materials and methods). This model has only three free parameters that need to be fitted from the data, representing the total concentration of each protein and the background growth in the PCA selection (see Materials and methods). In the model the changes in free energy ($\Delta\Delta G$, expressed in arbitrary units) for the mutations are additive but there is a sigmoidal relationship between PPI scores and $\Delta\Delta G$s (*Figure 3D*).

Fitting the three parameters from the data (*Figure 3—figure supplement 2A–B*) reveals that the model provides very good prediction of how mutations in the two proteins combine together

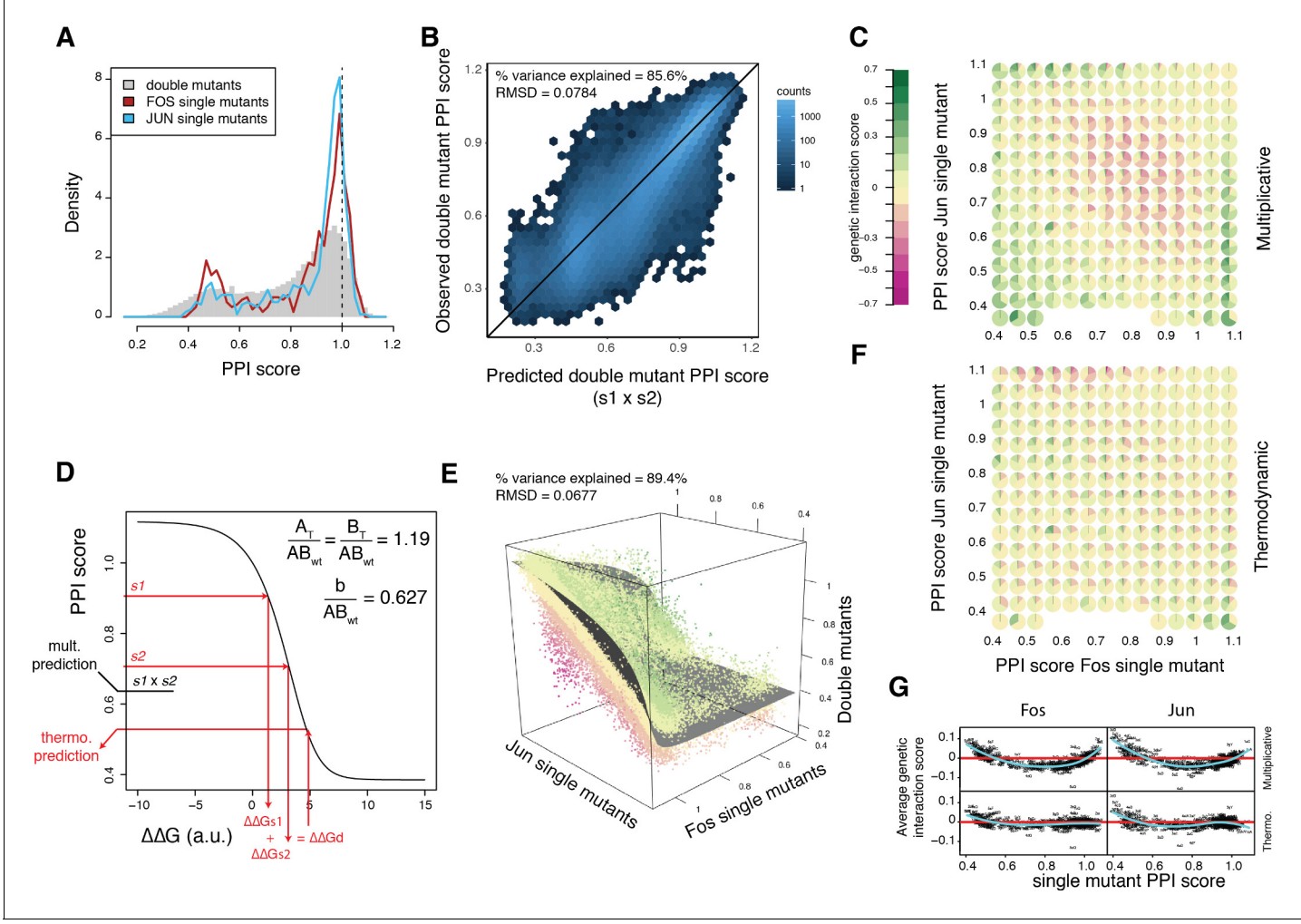

**Figure 3.** A thermodynamic model predicts double mutant outcomes. (**A**) Distribution of double mutants PPI scores compared to Fos and Jun single mutants. (**B**) Observed double mutants PPI scores against the scores predicted by a multiplicative model. (**C**) Pie chart array of genetic interaction score bins by Fos (*x-axis*) and Jun (*y-axis*) single mutant PPI score bins for genetic interactions calculated from the multiplicative model. (**D**) Fitted thermodynamic model (see Materials and methods). Red arrows illustrate how the sigmoidal function can lead to a different prediction than the multiplicative model. The three fitted free parameters are shown ($A_T/AB_{wt}$ and $B_T/AB_{wt}$ represent the total concentration of the two proteins relative to the concentration of the wild-type complex and $b/AB_{wt}$ represents the background growth relative to the concentration of the wild-type complex, see Materials and methods). $\Delta\Delta Gs1$, $\Delta\Delta Gs2$ and $\Delta\Delta Gd$ represent the change in free energy relative to the wild-type complex of the two single mutants and the double mutant, repsectively. (**E**) 3D scatter plot of double mutants PPI scores (*z-axis*) as a function of the corresponding Fos (*x-axis*) and Jun (*y-axis*) single mutants PPI scores with the fitted surface from the thermodynamic model. Dot color represents genetic interaction scores according to the color scale in (**C**). (**F**) Pie chart array of genetic interaction score bins by Fos (*x-axis*) and Jun (*y-axis*) single mutants PPI score bins for genetic interactions calculated from the thermodynamic model. (**G**) Average genetic interaction score across all double mutants involving a given Fos (*left*) or Jun (*right*) single mutant in function of its PPI score for the multiplicative (*top*) and thermodynamic (*bottom*) models. Data in all panels is for replicate 1.

DOI: https://doi.org/10.7554/eLife.32472.007

The following figure supplements are available for figure 3:

**Figure supplement 1.** Correlations between genetic interaction scores from the three biological replicates of the *trans* library selection.

DOI: https://doi.org/10.7554/eLife.32472.008

**Figure supplement 2.** Fitting the thermodynamic model on the *trans* library data.

DOI: https://doi.org/10.7554/eLife.32472.009

(percentage of variance explained of 89–90% in all three replicates, n = 107,618 mutation combinations, *Figure 3E*). The model also removes the systematic trend in the genetic interaction scores across mutation pairs with different individual effects (*Figure 3F–G*). Indeed, because of the sigmoidal nature of the model, a single mutant that decreases $\Delta\Delta G$ will increase PPI scores to a lower

extent in the wild-type context than when combined with a mutation that destabilized the complex because of the saturation effect caused by the plateau of the sigmoid.

## Specific interactions between structurally-related mutations

To investigate the remaining genetic interactions not accounted for by the thermodynamic model, we calculated residual genetic interaction scores as the difference between the observed double mutant PPI score and the thermodynamic model prediction. These new genetic interaction scores also correlate well amongst the three replicates, with a narrow peak centered on zero interaction and long tails of rare strong positive and negative genetic interactions (*Figure 3—figure supplement 2C*).

We observed more cases of strong negative (1711, 1.6%) than positive (883, 0.82%) genetic interactions (absolute score > 0.1, FDR = 0.2, *Figure 3—figure supplement 2D–G*). These strong interactions are enriched between particular Fos and Jun residues (*Figure 4—figure supplement 1*), with positive genetic interactions concentrated between positions close in the sequence of heptad positions (along the diagonal of the matrix in *Figure 4A*) and negative genetic interactions more spread-out in the structure and less enriched between specific pairs of positions (*Figure 4A–B* and *Figure 4—figure supplement 2*). Both directions of interaction are enriched between residues at the interface of the PPI and between residues close in space (*Figure 4C–D* and *Figure 4—figure supplement 3*), with this stronger for positive than for negative interactions. Positive interactions are therefore particularly enriched between contacting residues, identifying 'lock and key' specificity residues (*Horovitz, 1996*). We refer to these interactions beyond the interactions predictable from the global thermodynamic model as structural genetic interaction. For instance, in the wild-type PPI, the Glu residue in position 3g of Fos establishes a salt-bridge with the Arg residue in position 4e of Jun (*Figure 4E*). The individual mutations Glu3gLys and Arg4eGlu both destabilize the PPI by replacing the salt-bridge by repulsive electro-static interactions (Glu-Arg replaced by Lys-Arg and Glu-Glu with average PPI scores of 0.84 and 0.71, respectively). However, the two mutations compensate each other by recreating the salt-bridge (Glu-Arg replaced by Lys-Glu) and restoring a neutral PPI score of 0.98. Additional examples are shown in *Figure 4E*.

## Comparing genetic interactions in *cis* and *trans*

In addition to combining pairs of mutations in the two different proteins, we also quantified the effects of 17,688 double amino acid changes within Fos alone (99% of *cis* double mutant combinations reachable through combinations of single nucleotide changes; for all comparisons between the *trans* and *cis* libraries below, we only consider mutants reachable by single nucleotide changes in both libraries; *Figure 5A*, *Figure 5—figure supplement 1A* and *Supplementary file 3* and *4*). The PPI scores for single mutants correlate very well between the two libraries (R = 0.96, *Figure 5—figure supplement 1B*), further validating the reproducibility of the *deepPCA* method.

There is a good correlation between the PPI scores of *cis* and *trans* double mutants consisting of exactly the same pairs of substitutions at the same positions (R = 0.77, n = 5451 identical double mutants quantified in *cis* and *trans*, *Figure 5B*). This correlation indicates that the structural determinants of mutation effects in *FOS* and *JUN* remain well conserved despite sequence divergence over long evolutionary timescales. However, the distributions of double mutant effects are quite different for the *cis* and *trans* combinations (*Figure 5C*). This could be either due to different levels of genetic interactions or merely to the combination of different distribution of single mutant effects (*FOS* x *FOS* in *cis* and *FOS* x *JUN* in *trans*). To control for differences in the distributions of single mutant effects in the two libraries, we sub-sampled the libraries to match their single mutant effect distributions (*Figure 5—figure supplement 2A*, see Materials and methods). This revealed that, even when controlling for single mutant effect sizes, two mutations within Fos are more likely to increase the strength of the PPI than one mutation in Fos combined with a second mutation in Jun (p < $10^{-3}$ over 1000 sub-samplings, 5.2% vs 3.4%, respectively, for PPI scores > 1.04, *Figure 5—figure supplement 2B*). Two mutations in Fos are slightly less likely to destroy the PPI than a *trans* mutation combination (25.5% vs 27.8%, respectively, p < $10^{-3}$ for PPI scores < 0.64, *Figure 5—figure supplement 2C*) but have slightly more intermediate negative effects (p < $10^{-3}$, 39.4% vs 35.3%, respectively, for PPI scores between 0.64 and 0.92, *Figure 5—figure supplement 2D–E*). Whether mutations of the

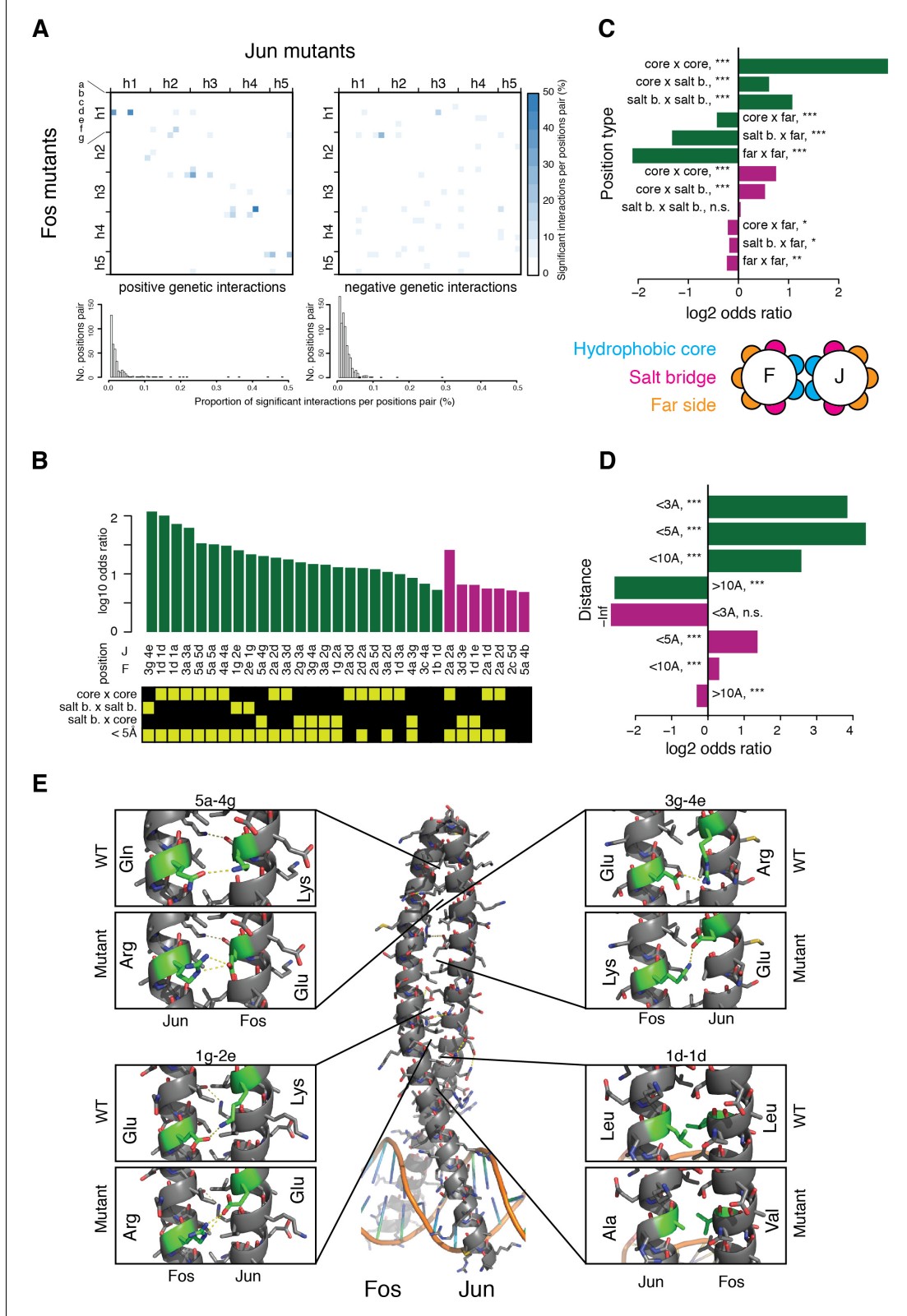

**Figure 4.** Structural genetic interactions. (**A**) Heatmap (*top*) and distribution (*bottom*) of percentage of significantly (absolute genetic interaction score > 0.1, FDR < 0.2) positive (*left*) or negative (*right*) genetic interactions per pairs of position. Pairs of positions without any significant interactions were excluded from the distribution (*bottom*). (**B**) Pairs of position significantly enriched in positive (*green*) or negative (*purple*) genetic interactions (Fisher's exact test, FDR = 10%). Each pair of position was classified according to its heptad position type and the distance between the two position

*Figure 4 continued on next page*

*Figure 4 continued*

(bottom matrix, yellow cells). (C–D) Enrichment for significantly positive (*green*) or negative (*purple*) genetic interactions between different position types (C) and at different distance threshold between the two positions (D). *, FDR < 0.1. **, FDR < 0.01. ***, FDR < 0.001. n.s., non-significant. (E) Example of structural interactions in the Fos-Jun complex (pdb: 1fos). Dashed yellow lines represent polar interactions predicted by pymol and the mutant structures were drawn using the pymol mutagenesis function.

DOI: https://doi.org/10.7554/eLife.32472.010

The following figure supplements are available for figure 4:

**Figure supplement 1.** Distribution of number of significant genetic interactions for Fos or Jun single mutants at different magnitude thresholds.

DOI: https://doi.org/10.7554/eLife.32472.011

**Figure supplement 2.** Robustness of enrichments in position pairs for significant (FDR < 0.2) genetic interactions at different magnitude thresholds.

DOI: https://doi.org/10.7554/eLife.32472.012

**Figure supplement 3.** Robustness of enrichments in structural features for significant (FDR < 0.2) genetic interactions at different magnitude thresholds.

DOI: https://doi.org/10.7554/eLife.32472.013

same individual effect sizes combine together in *cis* or in *trans* therefore influences the double mutant outcome.

## The thermodynamic model also accurately predicts interactions in *cis*

Because leucine zippers, including Fos and Jun, fold upon binding (*Patel et al., 1990*; *Thompson et al., 1993*), the same thermodynamic model based on a two-state equilibrium between the two unfolded proteins and the complex can describe how mutations combine in *cis* as well as in *trans*. We tested how well the thermodynamic model with parameters fitted on the *trans* double mutants predicted the *cis* library data and found that it gave very good prediction (percentage of variance explained of 82–83% for *cis* vs. 90–91% for *trans* combinations, *Figure 5—figure supplement 3A–B*). Similarly, fitting the thermodynamic model on the *cis* double mutants gave very good prediction of the *trans* library data (percentage of variance explained of 84% for *cis* and 90% for *trans*, *Figure 5—figure supplement 3A–B*). A common thermodynamic model therefore accounts very well for how mutations combine in both *cis* and *trans* to change the PPI (*Figure 5D* and *Figure 5—figure supplement 3C*). Therefore, just as in trans-, *cis*-genetic interactions have a component that results from the non-linear relationship between free energy and protein complex concentration.

## Structural interactions in the *cis* interaction landscape

We then tested whether the residual component of *cis*-genetic interactions are also enriched for structural interactions. The strongest cases of structural *cis* interactions (absolute genetic interaction score > 0.1 and FDR < 0.2 in both libraries, *Figure 5—figure supplement 4*) are indeed also enriched between proximally located residues but are less restricted to pairs of positions that are both at the PPI interface and involve more far side positions compared to *trans*-genetic interactions (*Figure 5—figure supplement 5*). *cis*-genetic interactions are also less enriched at specific positions and more dispersed throughout the structure (*Figure 5—figure supplement 6*). These results are robust to the magnitude threshold used to call strong genetic interactions and to the differences in single mutant effects between the two libraries (*Figure 5—figure supplement 7* and *Figure 5—figure supplement 8*). Thus, *cis*-genetic interactions can be subdivided into the same two components as *trans*-genetic interactions, genetic interactions that results from the non-linearity of the general relationship between protein complex concentration and free energy and specific structural interactions.

## Structural interactions are more abundant in *cis* than in *trans*

Interestingly, structural genetic interactions explain more of the variance in double mutant PPI scores when mutations are combined in *cis* than in *trans* (*Figure 5D*). Indeed, both positive and negative structural genetic interactions (genetic interactions not accounted for by the thermodynamic model; *Supplementary files 3* and *4*) are more abundant in *cis* than in *trans* (1493 vs. 835 true cases of positive and 1319 vs. 1128 true cases of negative genetic interactions in *cis* and *trans*, respectively, at p < 0.031, one sample t-test, FDR < 0.15 and 0.2; *Figure 5E* and *Figure 5—figure supplement 9*).

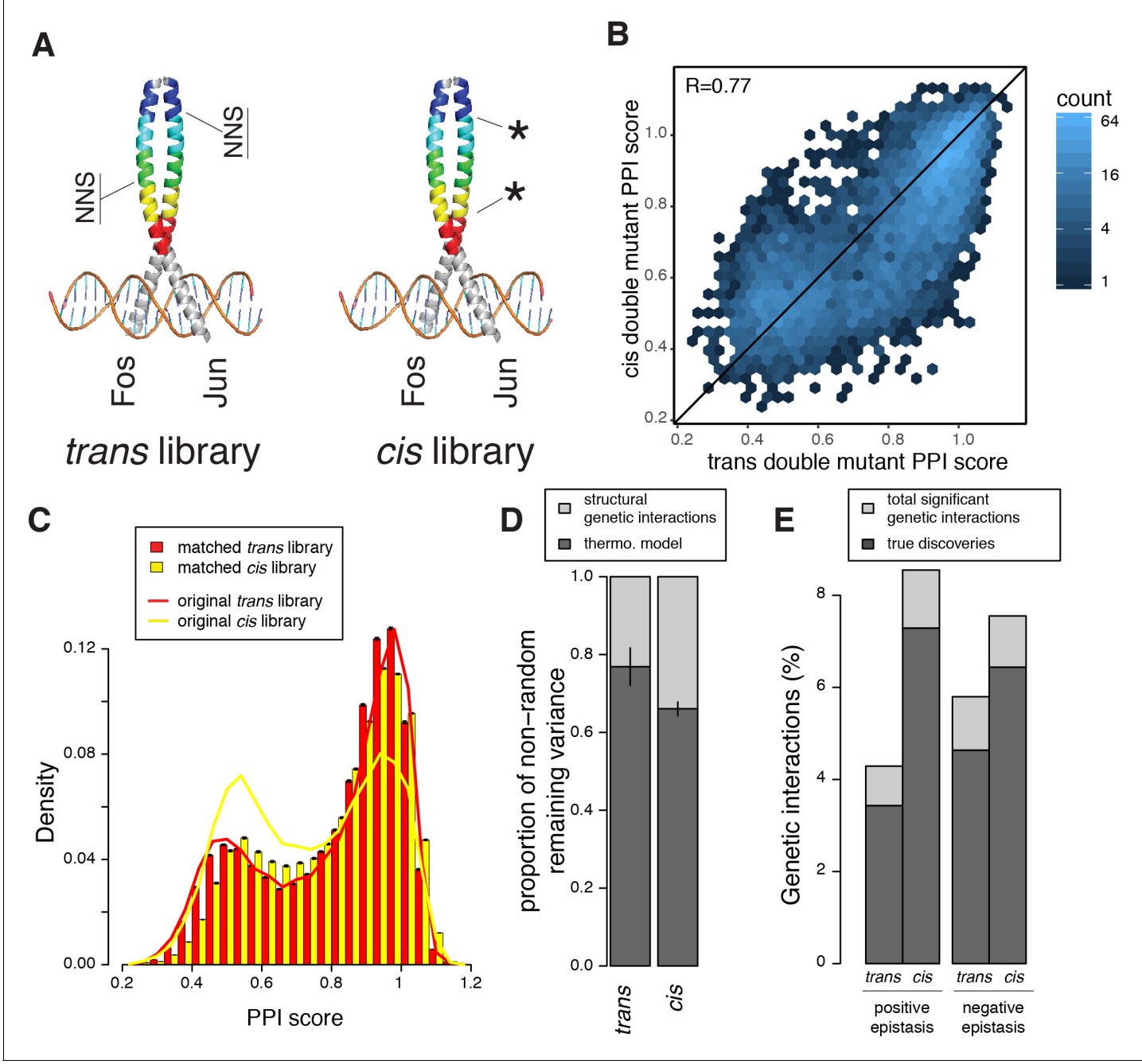

**Figure 5.** Comparison of double mutant mutation outcome and genetic interactions in *cis* and *trans*. (A) Cartoon illustrating how the *cis* library differs from the *trans* one. NNS, whole codon substitution. Asterisk, point mutation. (B) Scatter plot between Average PPI scores of identical pairs of mutations at the same positions in *cis* and *trans*. (C) Distribution of double mutants PPI scores in the original *cis* and *trans* libraries or after matching their single mutant effects distributions. Error bars represents 95% confidence intervals around the mean of 1000 sub-samplings when matching the two libraries. (D) Stacked bar-plot showing the proportion of the non-random variance in double mutant PPI scores that is not accounted for by the multiplicative model, explained by the thermodynamic model and the residual structural genetic interactions. Error bars represent the standard error of the mean. (E) Proportion of significant positive and negative genetic interactions in the two original libraries. See *Figure 5—figure supplements 2*, *3*, *7* and *9* for sub-sampled libraries with matched single mutant effect distributions.

DOI: https://doi.org/10.7554/eLife.32472.014

The following figure supplements are available for figure 5:

**Figure supplement 1.** PPI scores for the *cis* library.

DOI: https://doi.org/10.7554/eLife.32472.015

*Figure 5 continued*

**Figure supplement 2.** Proportion of *cis* and *trans* double mutants classified as strengthening, intermediate effect or severely detrimental using different score thresholds.
DOI: https://doi.org/10.7554/eLife.32472.016

**Figure supplement 3.** Fitting the thermodynamic model on both the *trans* and *cis* data.
DOI: https://doi.org/10.7554/eLife.32472.017

**Figure supplement 4.** Significant structural genetic interactions in *cis* and *trans*.
DOI: https://doi.org/10.7554/eLife.32472.018

**Figure supplement 5.** Comparisons of the patterns of structural genetic interactions in *cis* and *trans*.
DOI: https://doi.org/10.7554/eLife.32472.019

**Figure supplement 6.** Comparisons of the pairs of positions enriched in *cis* and *trans*.
DOI: https://doi.org/10.7554/eLife.32472.020

**Figure supplement 7.** Robustness of the enrichments in structural features when sub-sampling to match single mutant effects.
DOI: https://doi.org/10.7554/eLife.32472.021

**Figure supplement 8.** Robustness of enrichments in significant (FDR < 0.2) genetic interactions at different magnitude thresholds in the *cis* and *trans* libraries.
DOI: https://doi.org/10.7554/eLife.32472.022

**Figure supplement 9.** Comparisons of the extent of structural genetic interactions in *cis* and *trans*.
DOI: https://doi.org/10.7554/eLife.32472.023

This higher prevalence of genetic interactions in *cis* can potentially be explained by a higher number of contacts between positions within *FOS* than across the interface (1040 *cis* vs. 518 *trans* mutants pairs at positions within 5 Å of each other), supporting our previous result that proximity is a major determinant of genetic interactions both in *cis* and *trans* in a PPI interface.

## Discussion

Here, we have presented a protein-protein interaction assay – *deepPCA* – that allowed us to quantify the effects of >120,000 combinations of mutations in both *cis* and *trans* on the physical interaction between the products of the *FOS* and *JUN* proto-oncogenes. This provided a comprehensive and systematic data for how a very large number of different mutations in two genes combine to alter a biological function, allowing us to investigate the causes of genetic interactions between mutated genes and the extent to which genetic interactions can be quantitatively predicted. In its current form, *deepPCA* is limited to small domains because of the limit in the amplicon size for paired-end sequencing. However, the use of barcodes (*Hiatt et al., 2010*) would allow the assay to be applied to longer proteins.

Our data reveal that physical interactions in the cell generate two distinct types of genetic interaction: interactions due to the sigmoidal relationship between the concentration of a protein complex and the free energy of an interaction (*Figure 3*) and specific, structural interactions (*Figure 4*).

The general genetic interactions that arise in the physical interaction between molecules is one of several non-linear mappings that can occur between changes in genotype and changes in phenotype. Additional non-linearities occur in the folding of individual proteins or RNAs ('threshold robustness') (*Bershtein et al., 2006*; *Olson et al., 2014*; *Tokuriki and Tawfik, 2009*), in saturating enzyme flux (*Kacser and Burns, 1973*; *Stiffler et al., 2015*), and in regulatory dynamics (*Gjuvsland et al., 2007*; *Omholt et al., 2000*).

This type of genetic interaction is cumulative and easily predictable, for example a three parameter thermodynamic model accounts for ~90% of the variance in our dataset of >120,000 genotypes. The magnitude of this type of genetic interaction can also be predicted when combining three or more mutations together. A better knowledge of all the sources of non-linearities between the genotype and the phenotype is therefore critical to model how genotypic variation translates into phenotypic changes.

The second type of genetic interactions generated by molecular interactions is thermodynamically non-additive. These interactions are enriched between physically contacting and proximal residues, but can also involve some long-range indirect interactions (*Halabi et al., 2009*). Structural genetic interactions have a more complex basis and are therefore more difficult to predict. Gathering

comprehensive data similar to that described here for additional PPIs will help to further elucidate the structural determinants of genetic interactions and the rules for predicting them.

This second type of genetic interactions generated by a protein-protein interaction was more important when combining mutations in *cis* within the same protein than when combining mutations in *trans* between the two molecules. This results in a different distribution of double mutant outcomes when combining mutations in *cis* and *trans*. Whether a second mutation happens in *cis* or in *trans* can therefore impact an evolutionary outcome.

A substantial fraction of genetic interactions could however not be explained by structural contacts. Some other mechanisms not accounted for by the model could be at play. For instance, non-linearities between the growth rate and complementation complex in the protein-complementation assay could artificially produce genetic interactions. However, such saturation effects are unlikely in the range of expression and binding affinities in this study because they would lead to diminishing returns when combining two strength-increasing mutations, which is not observed (*Figure 3C*). Moreover, Levy et al. have shown that growth is correlated to complementation complex concentration over a wide-range of concentrations (*Levy et al., 2014*). A more likely source of actual genetic interactions could come from JUN's ability to form homodimers, which are however less stable than the Fos-JunN heterodimer (*Chinenov and Kerppola, 2001*). Mutations affecting the equilibrium between the Jun-Jun homodimer and the Fos-Jun heterodimer could indeed have effects that would not be predicted by our thermodynamic model. Elucidating the remaining mechanisms of genetic interactions will thus require further studies that take these effects into account.

Our approach complements the large-scale efforts to comprehensively map genetic interactions between gene deletions or representative alleles of yeast genes (*Tong et al., 2004*; *Costanzo et al., 2010*; *Costanzo et al., 2016*). Gene deletions are, however, rare in nature, with most genetic variation consisting of point mutations of diverse and difficult to predict effects. Our data provides a comprehensive view of how point mutations within two genes interact to affect a biological function. It will be interesting to extend this strategy to quantifying the effects of point mutation combinations on additional phenotypes beyond PPIs, including applying it to gene pairs that do not encode directly physically interacting proteins but instead participate in regulatory interactions or the same biological process.

## Materials and methods

All experiments were performed in triplicates starting from the transformation of the variant libraries into yeast.

Deep sequencing data is available at GEO with accession number GSE102901 (reviewer token: yvgbkwqajvypvah).

All perl and R scripts used to analyze the data are available at https://github.com/gdiss/Diss_et_al_eLife_2018 (*Diss, 2018*; copy archived at https://github.com/elifesciences-publications/Diss_et_al_eLife_2018)

### Yeast strain

All experiments were performed in BY4742 (MATα *his3Δ1 leu2Δ0 lys2Δ0* ura3Δ0).

### Media and buffer recipes

- LB: 10 g/L Bacto-tryptone, 5 g/L Yeast extract, 10 g/L NaCl. Autoclaved 20 min at 120°C.
- YPD: 20 g/L glucose, 20 g/L Peptone, 10 g/L Yeast extract. Autoclaved 20 min at 120°C.
- YPDA: YPD +40 mg/L adenine sulfate.
- SORB: 1 M sorbitol, 100 mM LiOAc, 10 mM Tris pH 8.0, 1 mM EDTA pH 8.0. Filter sterilized (0.2 µm Nylon membrane, ThermoScientific, Waltham, Massachusetts).
- Plate mixture: 40% PEG3350, 100 mM LiOAc, 10 mM Tris-HCl pH 8.0, 1 mM EDTA pH 8.0. Filter sterilized (0.2 µm Nylon membrane, ThermoScientific, Waltham, Massachusetts).
- Recovery medium: YPD +0.5 M sorbitol. Filter sterilized (0.2 µm Nylon membrane, ThermoScientific, Waltham, Massachusetts).
- SC -ura: 6.7 g/L Yeast Nitrogen base without amino acid, 20 g/L glucose, 0.77 g/L complete supplement mixture drop-out without uracile. Filter sterilized (0.2 µm Nylon membrane, ThermoScientific, Waltham, Massachusetts).

- SC –ura/ade/met: 6.7 g/L Yeast Nitrogen base without amino acid, 20 g/L glucose, 0.74 g/L complete supplement mixture drop-out without uracile, adenine and methionine. Filter sterilized (0.2 μm Nylon membrane, ThermoScientific, Waltham, Massachusetts).
- Competition medium: SC –ura/ade/met + 200 μg/mL methotrexate (BioShop Canada Inc., Canada), 2% DMSO.
- DNA extraction buffer: 2% Triton-X, 1% SDS, 100 mM NaCl, 10 mM Tris-HCl pH 8.0, 1 mM EDTA pH 8.0.

## Plasmid backbone construction

Two intermediate plasmids were constructed each carrying the *CYC* promoter, a DHFR F[1,2] (GGGGS)$_4$ linker fusion or *DHFR* F[3] (GGGGS)$_3$-(GGGAS) linker fusion, respectively, a CYC terminator and a *LEU2* or *URA3* selection cassette, respectively. The backbone fragments containing the *CYC* terminator, selection cassette and propagation elements were amplified from pAG415GPD-ccdB-EGFP and pAG416GPD-ccdB-EGFP (Addgene plasmid # 14196), respectively, by Polymerase Chain Reaction (PCR) using primer pairs OGD075-OGD077 and OGD076-OGD077, respectively. The *CYC* promoter fragment (identical for both constructs) was amplified by PCR from pYM-N10 (*Janke et al., 2004*) using primer pair OGD041-OGD042. The DHFR F[1,2] and DHFR F[3] fragments were amplified by PCR from pAG25-D1,2 and pAG32-D3 (*Tarassov et al., 2008*), respectively, using primer pairs OGD051-OGD080 and OGD053-OGD081, respectively. The linkers were encoded as overhangs on primers used to amplify *DHFR* fragments (3' side) and plasmid backbones (*CYC* terminator 5' side). All PCR fragments were purified using a QIAquick PCR purification kit (Qiagen, Netherlands). Plasmids were constructed by Gibson assembly using 15 fmol of backbone fragment, 45 fmol of *DHFR* fragment and 45 fmol of promoter fragment in 20 μL reactions. The reactions were incubated at 50°C for 60 min and 1 μL was transformed into homemade DH5α chemocompetent cells. Correct assembly was checked by colony PCR and the absence of mutations in the coding region by Sanger sequencing. The resulting sequence-confirmed plasmids were named pGD002 and pGD006, respectively.

The complete empty vector was then assembled by inserting the *CYC* promoter – DHFR F[1,2] – Linker fragment (amplified by PCR from pGD002 using primer pair OGD089-OGD101) into pGD006 (linearized by PCR using primer pair OGD087-OGD099) at the 3' end of the CYC terminator. All fragments were purified using a QIAquick PCR purification kit (Qiagen, Netherlands). Gibson assembly was performed as described above with 30 fmol of backbone and 90 fmol of fragment and transformed in homemade DH5α chemo-competent cells. Correct assembly was checked by colony PCR and the absence of mutations in the coding region by Sanger sequencing. The resulting sequence-confirmed plasmid was named pGD009.

## Cloning of FOS and JUN into the plasmid backbone

*FOS* and *JUN* basic leucine zipper domains, as defined by SMART (*Letunic et al., 2015*), were amplified from human cDNA provided by Juan Valcarcel using primer pairs OGD098-OGD110 and OGD113-OGD125, respectively. These fragments correspond to residues 135 to 199 and 250 to 314, respectively. The *CYC* terminator was amplified by PCR using primer pair OGD099-OGD135. The backbone was prepared by digestion with BamHI and NheI. All fragments were purified using a QIAquick PCR purification kit (Qiagen, Netherlands). Gibson assembly was performed as described above with 10 fmol of backbone and 50 fmol of each of the *CYC* terminator *FOS* and *JUN* fragments. Correct assembly was checked by colony PCR and the absence of mutations in the coding region by Sanger sequencing. The resulting sequence-confirmed plasmid was named pGD012.

## *trans* library

### Experimental design

Thirty-two codons in *FOS* and *JUN* were mutated using NNS primers (S = G or C), that is replacing the wild-type codon by one of 31 or 32 mutant codons (if the wild-type codon ends in A or T, all 32 codons on the primer are mutagenic) able to encode all 20 amino acids plus one stop. Assuming equal representation of all variants, replacing 32 positions by one of 32 codons leads to 1024 combinations. However, because 17/32 codons in wild-type *FOS* end in G or C, there are 1007 unique variant sequences each present at a frequency of 1/1024 and one wild-type sequence present at a frequency of 17/1024. Similarly, there are 24 codons in wild-type *JUN* ending in G or C. There are

thus 1000 unique variants each present at a frequency of 1/1024 and one wild-type sequence present at a frequency of 24/1024. Consequently, when combined, the resulting library design encodes 1,007,000 unique double codon mutants each present at a frequency of $1/1024^2$, 1007 unique single *FOS* codon mutants each present at a frequency of $24/1024^2$, 1000 unique single *JUN* codon mutants each present at a frequency of $17/1024^2$ and one wild-type sequence present at a frequency of $17 \times 24/1024^2$. The frequency of double amino acids mutants where both amino acids can be encoded by a single possible codon substitution each is thus equal to the frequency of double codon mutants, that is $f_{low} = 1/1024^2 = 9.5 \times 10^{-7}$. All the volumes for all reaction and culturing conditions, as well as the number of reads obtained through deep sequencing, were determined to ensure that no amino acid variant would be lost throughout the library construction process, the competition assay, the sequencing libraries preparation and the sequencing itself. Since some amino acids are encoded by a single codon, these frequencies where calculated using the codon as the unit. Additionally, the competition had to be run for a number of generations that had to be not too low, to ensure correct enrichment for strongly interacting variants, but also not too high, to ensure that the weakly interacting variants could still be quantified.

The mutagenic PCR steps caused no bottlenecks because the picomoles of mutagenic primers used in the reactions represented $\sim 1 \times 10^{12}$ molecules. The Gibson assembly was also performed with picomoles of *FOS* and *JUN* variants to ensure that inefficient assembly would not create a bottleneck. Following the recommendations of *Fowler et al. (2014)*, we targeted at least $10/f_{low} \approx 10^7$ bacterial and yeast transformants. Post-transformation selections (i.e. selection of transformants, before the actual competition) and competitions were performed in 1 L so that there were ~500 cells per variant when inoculating at an $OD_{600nm}$ of 0.05 (cell density of $10^7$ cells / OD/ mL x 0.05 OD x 1000 mL x $f_{low} \approx 500$). Two cycles of competition from an OD of 0.05 to 1.6 would thus lead to 10 population generations. An estimation of three to four generations of background growth (i.e. where weakly interacting variant grow as fast as strongly interacting variant at the beginning of the competition because of non-depleted cellular resources), leads to six to seven generations of effective competition. Under these settings, a non-functional variant (i.e. a variant that interact so weakly that the cell cannot grow) would decrease in frequency by $1/2^6$ to $1/2^7 \approx 64$ to 128 fold. A sequencing depth of $100 \times 1/f_{low} \approx 10^8$ reads per replicate input or output would thus lead to an average of 100 reads per variant in the input and ~1 read per dead variant in the output. A high number of reads is important to limit noise from stochastic sampling during cluster formation on the sequencing flow cell. However, as described below, the six samples libraries were sequenced on a single lane of an Illumina HiSeq 2500 for an average of $\sim 4.1 \times 10^7$ reads per sample, which was found to be enough for good quality data. The culture volume of 1 L also ensures that no bottleneck is created during DNA extraction (cell density of $10^7$ cells / OD/ mL * minimum of 1 OD x 1000 mL x $f_{low} \approx 10^4$ cells per variants). Finally, we needed to run a large number of PCR reactions per sample for the preparation of sequencing libraries to ensure that the libraries were amplified from enough template molecules. A lower number of template molecules would increase the probability of several reads coming from the same initial PCR template molecule hence biasing the measurement of variant frequency in the population. Targeting $10^8$ reads per sample, we decided to use 20 times more template molecules in the PCR reactions, that is a total of $2 \times 10^9$ per sample.

## Construction of the *trans* library by mutagenic overlap-extension PCR and Gibson assembly

Mutagenic NNS primers were ordered from Sigma-Aldrich. One forward and one reverse primer were designed per codon, with 15 nt upstream and 15 nt downstream of the target codon.

PCR1 template DNA fragments encompassing the whole coding sequences but lacking one or the other universal/flanking primer binding sites were prepared by restriction digestion in order to limit the amplification of wild-type sequences from the template DNA carried over from PCR1 to PCR2 by the two universal primers. All templates were prepared by digesting pGD012 with BsrGI and Cfr10I (*FOS* 5' template), BspHI (*JUN* 5' template) or BamHI and NheI (*FOS* and *JUN* 3' templates) and gel purified.

Since carried-over templates can still anneal and overlap-extend during PCR2, PCR1 was performed with a low template concentration. Specifically, two PCR reactions were set up per target codon, one using the 5' template, the reverse mutagenic primer corresponding to the target codon

and the 5' universal primer (OGD145 and OGD138 for FOS and JUN mutagenesis, respectively) and another one using the 3' template, the forward mutagenic primer corresponding to the target codon and the 3' universal primer (OGD137 and OGD148, respectively). PCR reactions were performed with Kapa HiFi HotStart DNA polymerase (Kapa Biosystems, Wilmington, Massachusetts) according to manufacturer's protocol in 25 μL with 0.15 fmol of template DNA and 7.5 pmol of each primer and using a melting temperature of 58°C and an extension time of 1 min for 15 cycles. This resulted in two PCR fragments per target codon that overlap over 33 bp. The 5' fragment started 169 or 142 bp upstream of the *CYC* promoter (for *FOS* and *JUN*, respectively) and encompassed the *CYC* promoter, *DHFR* fragment, linker and first half of the gene until 15 bp downstream of the target codon. The 3' fragments started 15 bp upstream of the target codon and encompassed the rest of gene and a fragment of the CYC terminator (the 3' and 5' fragments for *FOS* and *JUN*, respectively, overlapping over 40 bp for Gibson assembly).

The two overlapping PCR products of each target codon were then pooled together (20 μL each) and purified using QIAEX II beads as follows. 120 μL of QX1 buffer, 1.5 μL of beads and 10 μL of 3 M NaOAc were added to the 40 μL of PCR mix. The mixtures were vortexed, incubated 10 min at room temperature and spun for 2 min at 3184 *g* (max speed) in an Eppendorf Centrifuge 5810 R. The beads were washed twice with 100 μL of buffer PE and the DNA fragments were eluted in 30 μL buffer EB after a 45 min room temperature incubation.

The second PCRs were performed with Kapa HiFi HotStart DNA polymerase (Kapa Biosystems, Wilmington, Massachusetts) according to manufacturer's protocol in 25 μL with 10 μL of the eluted DNA fragments, 7.5 pmol of each universal primer (OGD137-OGD145 or OGD148-OGD138 for *FOS* and *JUN*, respectively) and using a melting temperature of 53°C and an extension time of 1 min 30 s for 15 cycles. 3 μL of the PCR product of each target codon were analyzed on an agarose gel to ensure the correct overlap extension. The concentrations of the PCR products of correct size were estimated by four replicate measurements for each target codon using Quant-iT Pico-Green dsDNA Assay Kit (Life Technologies, Carlsbad, California) according to manufacturer's protocol and corrected by the intensity ratio of the correct band over the whole well measured on the agarose gel. The 32 PCR products corresponding to the same gene were then pooled at equimolar ratio, resulting in two libraries of mutated PCR products at target codon positions, one for *FOS* and one for *JUN*.

The backbone was amplified by PCR using Q5 Hot Start High-Fidelity DNA Polymerase (New England Biolabs, Ipswich, Massachusetts) according to the manufacturer's protocol in 8 × 50 μL with 2 ng of pGD012 template DNA and 22.5 pmol of each primer (OGD144 and OGD153) per 50 μL reaction and using a melting temperature of 58°C and an extension time of 2 min 30 s for 24 cycles.

The two mutated PCR product libraries overlapped each other over 40 bp in the CYC terminator. They also each overlapped upstream of the promoter with the backbone over 40 bp. These Gibson assembly junctions were chosen to minimize the effect of indels on protein activity that can result from incorrect assembly. All three DNA fragments were gel purified using the QIAquick gel extraction kit (Qiagen, Netherlands) and dialyzed to remove extra salt using nitrocellulose membranes with 0.025 μm pores (Merck Millipore). The three fragments were assembled by Gibson assembly in 800 μL with 1.2 pmol of backbone fragment and 3 pmol of each of the mutated gene fragments. The master mix was split into 16 × 50 μL reactions and incubated 17 hr at 50°C. The 16 reactions were then pooled back, purified on column using QIAquick PCR purification kit (Qiagen, Netherlands), eluted in 2 × 30 μL of EB buffer, concentrated down to 29 μL in a speedvac and stored at −20°C until use.

Test transformations revealed that a 2X dilution gave the highest number of transformants. 5 μL of concentrated DNA assembly were thus diluted with 5 μL of distilled water and transformed in two batches into NEB 10-beta Electrocompetent *E. coli* as follows. For one batch, four transformations were set up by mixing 1 μL of DNA to 25 μL of competent cells in a 0.1 cm Gene Pulser Cuvette (Bio-Rad, Hercules, California) chilled on ice. The four mixtures were electroporated using a Gene Pulser Xcell (Bio-Rad, Hercules, California) at 2.0 kV, 200 Ω and 25 μF. After each electroporation, the cells were immediately re-suspended in 1 mL SOC medium (provided by the manufacturer) warmed to 37°C and the cuvettes were rinsed with another 1 mL of the same medium. The four electroporations were then pooled together and incubated for 30 min at 37°C. After this recovery period, 10 μL were diluted 100 X in the same medium and 10 μL were plated on pre-warmed LB +Amp Petri dishes and incubated for ~16 hr at 37°C. The remaining ~7.7 mL of cells were added

to 200 mL of pre-warmed LB +400 µg/mL in 1L flasks and incubated for ~16 hr at 37℃ under constant agitation at 200 rpm. This resulted in a total of 11.6e6 transformant per batch. Each batch of 200 mL was then split in four 50 mL Falcon tubes, cells were harvested by centrifugation for 15 min at 3184 $g$ (max speed) in an Eppendorf Centrifuge 5810 R and pellets were stored at −20℃ for later use. One pellet of each batch was then thawed and plasmid DNA was extracted using Qiagen Plasmid Midi Kit. The two batches were then pooled at equimolar ratio to form the plasmid DNA library.

## Large-scale yeast transformation

Yeast transformation was performed in triplicates in parallel following a modified version of the protocol found in *Melamed et al. (2013)*. First, a glycerol stock of the BY4742 strain was streaked to single colonies on YPD Petri dishes and incubated at 30℃ for 2 days. For each replicate, a whole colony was resuspended in 25 mL YPDA and grown overnight to saturation at 30℃ under constant agitation at 200 rpm. The next day, three flasks of 700 mL of pre-warmed YPDA were inoculated with each of the pre-culture at a starting $OD_{600nm}$ of 0.125 and incubated at 30℃ under constant agitation at 200 rpm. After 6 hr 30 min of incubation, the cells were harvested by centrifuging 5 min at 3000 $g$ in a Avanti J-26 XP Centrifuge using a JLA 8.1000 rotor (Beckman-Coulter, Brea, California). Cells were washed once with 250 mL sterile distilled water, re-suspended in 25 mL SORB, transferred to 50 mL Falcon tubes and completed to 50 mL with SORB. The cells were harvested by centrifugation 5 min at 3000 $g$ in an Eppendorf Centrifuge 5810 R, re-suspended in 34.3 mL SORB and incubated at room temperature on a wheel for 30 min. Meanwhile, 2.2 mL of 10 mg/mL salmon sperm DNA (Agilent Technologies, Santa Clara, California) were thawed, boiled for 5 min in a heating block and cooled on ice for 2 min. When the cells were ready, 56 µg of plasmid DNA library and 700 µL salmon sperm DNA were added to each replicate and mixed by inverting the tubes six times. Each *trans* replicate was then split in 4 × 8.75 mL in 50 mL Falcon tubes for a total of 12 tubes, to which 35 mL of plate mixture were added. The Falcon tubes were incubated at room temperature for 30 min on a SSM4 mini seesaw rocker (Stuart Equipment, United Kingdom) at 30 oscillations/min. Then, 3.5 mL of DMSO were added to each tube and mixed in by inverting the tubes six times. The cells were heat-shocked for 20 min at 42℃ in a water bath. To homogenize the temperature faster, the tubes were inverted after 30 s, 1 min, 2 min 30 s, 5 min, 7 min 30 s, 10 min and 15 min. The cells were then harvested by centrifugation for 5 min at 1791 $g$ (3000 rpm) in an Eppendorf Centrifuge 5810 R (Eppendorf, Hamburg, Germany). The supernatant was first poured out of the tubes, the tubes were then quick-spun for 10 s to bring the remaining liquid down, which was then removed by aspiration with a vacuum pump. The cells from each tube were re-suspended in 10 mL of pre-warmed recovery medium. The four tubes of the same replicate were pooled together and added to 1.4 L of pre-warmed recovery medium in 5 L flasks. Recovery medium were incubated for 1 hr at 30℃ under constant agitation at 200 rpm. The cells were then harvested by centrifugation at 3000 $g$ for 5 min in a Avanti J-26 XP Centrifuge using a JLA 8.1000 rotor (Beckman-Coulter, Brea, California), washed once with 200 mL of SC –ura and added to 1.4 L of SC –ura in 5 L flasks. After homogenization by stirring, 10 µL were platted on SC –ura Petri dishes and incubated for ~48 hr at 30℃ to measure transformation efficiency. The liquid cultures were grown to saturation for ~48 hr at 30℃ under constant agitation at 200 rpm. This transformation resulted in 19.88 × 10⁶, 19.32 × 10⁶ and 16.52 × 10⁶ transformants for replicates 1, 2 and 3.

## Competition assay

After the first cycle of post-transformation selection, a second selection cycle was performed for each replicate by inoculating 1 L of SC –ura/ade/met at a starting $OD_{600nm}$ of 0.1 with the saturated culture and growing it to exponential phase ($OD_{600nm}$ of ~1–1.2) for 12 hr 30 min at 30℃ in 5 L flasks under constant agitation at 200 rpm. To start the first cycle of the competition, the volume of culture necessary to inoculate 1 L of competition medium at an $OD_{600nm}$ of 0.05 was centrifuged for 5 min at 3,000 $g$ in an Eppendorf Centrifuge 5810 R. Cells were re-suspended in 10 mL of competition medium containing methotrexate and added to 1 L of the same medium and incubated at 30℃ in 5 L flasks under constant agitation at 200 rpm. Meanwhile, the remainder of the cultures were harvested by centrifugation for 5 min at 3,000 $g$ in a Avanti J-26 XP Centrifuge using a JLA 8.1000 rotor (Beckman-Coulter, Brea, California), washed with distilled water, transferred to 50 mL Falcon tubes and the pellets were stored at −20 for later use. After ~20 hr 50 min of growth, the three replicates

reached an $OD_{600nm}$ of 1.57, 1.60 and 1.53 respectively. A second cycle of competition was performed in the same conditions by diluting back the cultures to an $OD_{600nm}$ of 0.05 in 1L of competition medium. The remainder of the cultures were harvested and stored as described above. After 16 hr of growth, the cultures reached an $OD_{600nm}$ of 2.66, 2.49 and 2.55, respectively. The whole cultures were harvested and stored as described above.

## DNA extraction

The cell pellets of the second post-transformation selection cycle (i.e. input population) and second competition cycle (i.e. output population) were first thawed at room temperature, re-suspended in 10 mL of DNA extraction buffer, frozen for 10 min in an ethanol/dry ice bath and thawed for 10 min in a 62 water bath. After an additional freeze/thaw cycle, 10 g of acid-washed, 425–600 µm glass beads (Sigma-Aldrich, St. Louis, Missouri) and 10 mL of Phenol/Chloroform/Isoamyl alcohol 25:24:1 equilibrated in 10 mM Tris-HCl, 1 mM EDTA, pH 8.0 were added to each tube. Cells were lysed by vortexing for 10 min and the aqueous phase containing the total DNA was separated by centrifuging for 20 min at 3,184 $g$ (max speed) in an Eppendorf Centrifuge 5810 R (Eppendorf, Hamburg, Germany). Another extraction was performed by adding 11 mL of Phenol/Chloroform/Isoamyl alcohol 25:24:1 equilibrated in 10 mM Tris-HCl, 1 mM EDTA, pH 8.0, vortexing 2 min and centrifuging 20 min at 3,184 $g$ (max speed) in an Eppendorf Centrifuge 5810 R (Eppendorf, Germany). The aqueous phase was transferred to a new tube and nucleic acids were precipitated by adding 1/10 vol of 2M NaCl and 2.2 volumes of pre-chilled absolute ethanol and incubating for 30 min at −20℃. NaCl was used instead of NaOAc to limit SDS precipitation. Nucleic acids were pelleted by centrifugation for 30 min at 3,184 $g$ (max speed) in an Eppendorf Centrifuge 5810 R (Eppendorf, Germany) pre-chilled at 4℃. The supernatants were removed and the pellets were put at 37℃ until dry and then re-dissolved in 6 mL of 10 mM Tris-HCl, 1 mM EDTA, pH 8.0. The dissolution was extremely slow, and after an overnight incubation at room temperature, the pellets were still not completely dissolved. The solution was thus centrifuged for 15 min at 3184 $g$ (max speed) in an Eppendorf Centrifuge 5810 R (Eppendorf, Germany) to pellet the non-dissolvable material and the supernatants were transferred to new 14 mL Falcon tubes. RNAs were digested by adding 50 µL of RNAseA 10 mg/ml and incubating for 30 min at 37℃ followed by a 2 hr incubation at room temperature. DNA was then extracted by adding 6 mL of Phenol/Chloroform/Isoamyl alcohol 25:24:1 equilibrated in 10 mM Tris-HCl, 1 mM EDTA, pH 8.0. After centrifugation 10 min at 3184 $g$ (max speed) in an Eppendorf Centrifuge 5810 R, the interphases were still thick and two other Phenol/Chloroform/Isoamyl alcohol 25:24:1 extractions were performed in the same way followed by an extraction with chloroform alone. DNA was precipitated by adding 600 µL of 2M NaCl and 13.2 mL of pre-chilled absolute ethanol and incubating for 20 min at −20℃ and then pelleted by centrifugation 30 min at 3184 $g$ (max speed) in an Eppendorf Centrifuge 5810 R (Eppendorf, Germany) pre-chilled at 4℃. To remove excess salt, 8 mL of 70% ethanol was poured on top of the pellets and incubated for 30 min at room temperature. After centrifugation 30 min at 3184 $g$ (max speed) in an Eppendorf Centrifuge 5810 R (Eppendorf, Germany), the supernatant was removed, the tubes were sealed with an air-breathable seal and the pellets were let to dry overnight at room temperature. The next day, the dried DNA pellets were dissolved in 1 mL of 10 mM Tris-HCl, pH 8.0. The three replicates from competition cycle two appeared to be cloudy. To remove the non-dissoluble material, the six samples were centrifuged 5 min at 3184 $g$ (max speed) in an Eppendorf Centrifuge 5810 R (Eppendorf, Germany) and the supernatants were transferred to 1.5 mL Eppendorf tubes (Eppendorf, Germany).

## Sequencing library preparation

The sequencing libraries were constructed by two consecutive PCR reactions using a method adapted from *Levy et al. (2015)*. The first PCR was designed to amplify the region of interest, that is from directly upstream of the first mutated codon in *FOS* to directly upstream of the first mutated codon in *JUN* (the two genes are in head-to-tail orientation). The first PCR also added Unique Molecular Barcodes (UMIs) and the first half of the illumina adapter sequences. A small number of cycles of the first PCR would limit the number of differently barcoded molecules that derive from the same template molecule. The second PCR would then add the remainder of the Illumina adapter sequences.

Plasmid concentrations in the total DNA extractions were first quantified by qPCR using primer pair OGD241-OGD242 that bind in *ori* region of the plasmid. For each six samples of each of the *cis* and *trans* libraries, respectively, 4 and 48 PCRs were performed using Q5 Hot Start High-Fidelity DNA Polymerase (New England Biolabs, Ipswich, Massachusetts) according to manufacturer's protocol in 50 µL reactions with $4.2 \times 10^7$ molecules of plasmid from the DNA extraction, 25 pmol of primers OGD237 and OGD238, and a melting temperature of 66°C (previously determined by temperature gradient), an extension time of 30 s or 1 min, respectively, for four cycles. A total of $2 \times 10^9$ molecules of plasmids were then used to prepare the sequencing libraries from each sample. Excess primers were removed by adding 2 µL of ExoSAP-IT (Affymetrix, Santa Clara, California) and incubating for 20 min at 37°C followed by an inactivation for 15 min at 80°C. This step was necessary because these 60 nt primers are not efficiently removed by the following column purification step. The 48 PCRs of each sample of the *trans* library were then pooled and purified using eight Qiagen PCR purification kit (Qiagen, Netherlands) columns per sample. According to manufacturer's protocol, one column is able to bind 10 µg of DNA, which corresponds to $\sim 8 \times 10^8$ genomes. Eight columns were then used to ensure that they are not saturated by the genomic DNA carried over from the DNA extraction. The DNA was eluted in $2 \times 50$ µL of EB buffer (provided by the manufacturer) and pooled for each sample. The eluted DNA was then split into 24 PCR reactions per, which were performed using Kapa HiFi HotStart DNA polymerase (Kapa Biosystems, , Wilmington, Massachusetts) according to manufacturer's protocol in 50 µL with 15 pmol of illumina adapter primers. The reverse primers carried a different index for each of the six samples of the same library. For this PCR step, Kapa was chosen over Q5 because it was less efficient in the first PCR reactions (higher optimal melting temperature) and thus would lead to a lower re-barcoding of amplicons with new UMIs present on primers from the first PCR reaction that would have been carried over. Each sample was loaded on an agarose gel to check for correct amplification. A strong non-specific band of lower size was observed, which seemed to gradually disappear as the number of cycles in the first PCR was increased. However, increasing the number of cycles would increase the probability of producing amplicons with different UMIs that derived from the same original template molecule. The number of cycles in the first PCR was thus kept to four and the band of correct size was extracted by gel purification from each sample. To this end, the 24 PCRs of each sample were pooled, concentrated on four Qiagen PCR purification kit (Qiagen, Netherlands) columns per sample and each eluted with $2 \times 50$ µL of EB buffer (provided by the manufacturer). The bands of correct size were then purified on 2% agarose gel starting from 100 µL of each sample using 10 µL QIAEX II beads (Qiagen, Netherlands) according to manufacturer's protocol and eluted in 20 µL EB buffer. DNA concentration was determined by picogreen in triplicates and the six samples were pooled at equimolar ratio. The pooled sample was sequenced in a single lane of an Illumina HiSeq2500 with 125 bp paired-end reads at the EMBL Genomics Core Facilities in Heidelberg, Germany.

## Sequencing data analysis and filtering

A total of 245,676,608 read pairs were obtained from the lane. However, ~30% corresponded to adaptor sequences and had to be discarded. The high adaptor sequence reads level was probably due to the non-specific PCR amplification during the sequencing library preparation despite the gel purification. Sequencing data was filtered using homemade Perl scripts. Paired reads for which one of the two variable regions had an average Phred score below or equal to 20 were also discarded. Additionally, paired reads were also filtered-out if they had i) one or more non-resolved bases (Ns) in the variable or UMI regions, ii) more than one mutated codon in each gene's variable region or iii) if the mutated codon ended in an A or T (since these were not encoded by the NNS mutagenic primers). These non-designed substitutions could have occurred during the library construction process before the competition and thus lead to accurate PPI score. However, their low frequency suggested they were more likely to be the results of amplification errors during sequencing library preparations and/or sequencing errors and were thus filtered-out in the interest of stringency and higher quality. These filtering steps resulted in between 1.08 and $1.42 \times 10^7$ usable reads per replicate input and output.

To estimate sequencing errors, we measured the per base error probability in the constant region upstream of the variable regions of *FOS* and *JUN* that were used as primer annealing sites for library amplification. These sequences should be 100% wild type and any mutations called should be the

result of a sequencing error. We estimate the probability of a base to be wrongly called to be 0.0019 for the forward read and 0.0058 for the reverse read, averaged across the six samples (three replicates inputs and outputs). Sequencing errors do not seem to affect the downstream estimate of PPI scores as filtering reads at a Phred score of 30, 35 or additionally filtering out reads where at least one of the called mutations had a Phred score below 30 (>90% of the wrongly called mutations in the constant region had a Phred score below 30 compared to <25% of the correctly called bases) led to PPI scores that were highly correlated to the ones calculated using the above filters (*Figure 1—figure supplement 1A*).

## PPI score calculation and correction for background growth

The frequency of each variant in each sample was then calculated by counting the number of unique UMIs associated to each unique amino acid variant (wild-type (WT), single or double mutant) divided by the total number of unique UMI per variant in the corresponding sample. First, nucleotide sequences were converted to amino acid sequences and the UMIs of different nucleotide variants coding for the same amino acid sequences were summed up. PPI scores were then calculated for each variant according to the formula provided in *Figure 1B*.

Variants with zero UMI in any of the three outputs were filtered out. Even though variants with lethal mutations could be expected to go to 0, they cannot go further down while their actual frequency in the population is still decreasing (e.g. they could still not be detected even if sequencing 10X or 100X deeper). Their PPI scores thus reach a boundary that is a function of the input UMI count as illustrated in *Figure 1—figure supplement 1B*. These variants could be misidentified as having intermediate effects or even neutral effects while they are actually lethal. On the contrary, variants with at least one UMI in each replicate output have been consistently detected. Hence, even though the estimate of their PPI score might be associated to a high error due to the stochasticity of low read counts, their actual PPI score should lie within a range around the estimates (e.g. sequencing 10X or 100X deeper would lead to read counts of ~10 or ~100, respectively).

Variants with low input UMIs are also typically associated with a higher error due to stochastic effects. Moreover, these variants are biased toward intermediate or neutral effect due to the PPI score boundary (*Figure 1—figure supplement 1B*). We thus filtered out variants with less than 10 UMIs in any of the three input replicates. The WT and variants containing stop codons were also excluded from downstream analyses.

In PCA, cells that do not express the DHFR fragments or express protein fusions that do not interact can initially divide for a few generations. This background growth is probably the result of intracellular stores of folate and other metabolites that can be used for growth before they are depleted. To correct for batch effects in this background growth, PPI scores were corrected to align the mode of the detrimental variants (left most mode in the bimodal distribution of PPI scores), which is assumed to correspond to the background growth. This mode was identified in each replicate using the R *density* function. PPI scores were then aligned to the maximal value of this mode across replicates to avoid the possibility of getting negative values for variants with PPI scores in the left-hand tail of the distribution. This was done according to the following formula, which keeps the WT PPI score as one in all replicates:

$$corrected\ PPIscore = \frac{PPIscore - \frac{m-M}{1-M}}{PPIscore_{wt} - \frac{m-M}{1-M}}$$

where $m$ is the mode in the corresponding replicate and $M$ is the maximal value of the modes across all replicates.

## Small-scale confirmation of single mutant effects

To ensure that growth rate measured by deep sequencing in our assay agree with actual growth rate, we reconstructed 14 variants (one wild-type interaction and 13 single mutants) chosen randomly across the range of PPI scores and measured their growth rate individually in a TECAN plate reader (TECAN, Switzerland). The 14 mutants were constructed individually by OE-PCR in the same way as the large-scale library but using fixed sequence mutagenic primers instead of NNS primers (*Supplementary file 5*). Each clone was sequence-confirmed and transformed in yeast using a scaled-down version of the transformation protocol used for the large-scale assay.

Two colonies of each variant were grown overnight in 100 μL of SC –ura/ade/met medium at 30C in microtiter plate with constant shaking. The next day, precultures were diluted to an $OD_{600nm}$ of 1 and 5 uL was added to a final volume of 100 μL of MTX medium in a microtiter plate. Growth curve were measured in a TECAN infinite M200 (TECAN, Switzerland) with $OD_{600nm}$ measurements every 15 min for 50 hr.

PPI scores were calculated in a similar way than the ones from the large-scale assay. For each variant, the number of doubling was calculated between the beginning of the assay and the time point at which the wild-type interaction reached its maximal growth rate (approximating the point at which cells were harvested in the large-scale assay). The PPI score was then calculated as the number of doublings of a variant relative to the number of doubling of the wild-type.

## Classification of mutant effects

Thresholds based on the distributions of PPI scores averaged across replicates were used to classify variants as detrimental (PPI score <= 0.64), intermediate (0.64 < PPI scores<=0.96), neutral (0.96 < PPI scores<=1.04) or strengthening (PPI scores > 1.04). Additionally, one sample t-tests were performed to test the difference of the mean PPI score of each variant across replicates from the WT PPI score of 1. False discovery rates were controlled using the Benjamini-Hochberg method as implemented in the R function *p.adjust*. The p-value threshold to call variants significantly different from WT was set so that the FDR was inferior to 0.05. The robustness of all downstream analyses was analyzed by varying these PPI score thresholds.

## Prediction of single mutant effects from amino acid physic-chemical features

Fos and Jun single mutant effects were predicted at each of the 32 mutated positions from amino acid features retrieved from the amino acid index database (*Kawashima et al., 2008*). Only features without any missing value for the 20 amino acids were considered. PPI score of mutations at each position were predicted with each feature independently using R's built-in *lm* function.

## Comparisons of identical substitutions in Fos and Jun

The differences in PPI scores between identical mutations at the same position in Fos and Jun were assessed using a paired t-test. False discovery rates were controlled using the Benjamini-Hochberg method as implemented in the R function *p.adjust*. The enrichment for mutations that had an intermediate effect in either protein was tested using Fisher's exact test.

## Multiplicative genetic interactions

Multiplicative genetic interactions were computed as the difference between the PPI scores measured for the double mutant and the multiplicative expectation, that is the product of the PPI scores of the two corresponding single mutants. This was done independently in each of the three replicates, and an average genetic interaction score was calculated for each variant. A one-sample t-test was performed to test whether this average was significantly different from 0. FDRs were computed as follows. In each of 10,000 permutations, the genetic interaction scores were randomized within replicates independently. The one-sample t-test was then performed again. At any given p-value threshold, hits in the permuted data can be considered as false discoveries that are significant just by chance by randomly drawing values from this distribution of genetic interaction scores. At each p-value observed in the original data, the FDR was hence computed in each permuted data set as the number of false discoveries divided by the total number of discoveries in the real data. The standard error of the average FDR was then computed in the same way at each p-value threshold, but was negligible ($2.5e^{-5}$ at a FDR of 0.2).

## Thermodynamic model

The thermodynamic model was built by fitting the quadratic solution for the dissociation constant, $K_D$, of the interaction:

$$e^{\frac{\Delta G}{RT}} = K_D = \frac{A_F \times B_F}{AB} = \frac{(A_T - AB) \times (B_T - AB)}{AB} \qquad (1)$$

with $AB$ being the concentration of the dimeric complex, $A_F$ the concentration of the free form of

protein A, $B_F$ the concentration of the free form of protein B, $A_T$ the total concentration of protein A and $B_T$ the total concentration of protein B.

Assuming that the growth rate of a cell expressing a variant is directly proportional to the concentration of protein complex so that

$$n = AB \tag{2}$$

with $n$ being the number of generations during competition. Cells with variants harboring lethal mutations such as stops are still able to grow for a few generations at the beginning of the competition, even in the presence of methotrexate, as highlighted by the mode of the peak of detrimental variants in *Figure 3A*. This background growth might be allowed before the depletion of intracellular stores of tetrahydrofolate or downstream metabolites. We hence introduced a parameter $b$ to account for background growth. PPI score (*PPI*) can then be expressed as

$$PPI = \frac{n+b}{n_{wt}+b} = \frac{AB+b}{AB_{wt}+b} \quad \rightarrow \quad AB = PPI \times AB_{wt} + b \times (PPI - 1) \tag{3}$$

Replacing (3) in (1),

$$e^{\frac{\Delta G}{RT}} = \frac{(A_T - PPI \times AB_{wt} - b \times (PPI-1)) \times (B_T - PPI \times AB_{wt} - b \times (PPI-1))}{PPI \times AB_{wt} + b \times (PPI-1)}$$

$$= \frac{AB_{wt} \times \left(\frac{A_T}{AB_{wt}} - PPI - \frac{b}{AB_{wt}} \times (PPI-1)\right) \times \left(\frac{B_T}{AB_{wt}} - PPI - \frac{b}{AB_{wt}} \times (PPI-1)\right)}{PPI + \frac{b}{AB_{wt}} \times (PPI-1)}$$

$$= \frac{AB_{wt} \times \left(\frac{A_T}{AB_{wt}} + \frac{b}{AB_{wt}} - PPI \times \left(\frac{b}{AB_{wt}} + 1\right)\right) \times \left(\frac{B_T}{AB_{wt}} + \frac{b}{AB_{wt}} - PPI \times \left(\frac{b}{AB_{wt}} + 1\right)\right)}{PPI \times \left(\frac{b}{AB_{wt}} + 1\right) - \frac{b}{AB_{wt}}}$$

$$= \frac{AB_{wt} \times \left(\frac{\frac{b}{AB_{wt}} + \frac{A_T}{AB_{wt}}}{\frac{b}{AB_{wt}} + 1} - PPI\right) \times \left(\frac{\frac{b}{AB_{wt}} + \frac{B_T}{AB_{wt}}}{\frac{b}{AB_{wt}} + 1} - PPI\right)}{PPI \times \left(\frac{b}{AB_{wt}} + 1\right) - \frac{b}{AB_{wt}}}$$

$$\Delta G = RT \times log\left(\frac{AB_{wt} \times \left(\frac{\frac{b}{AB_{wt}} + \frac{A_T}{AB_{wt}}}{\frac{b}{AB_{wt}} + 1} - PPI\right) \times \left(\frac{\frac{b}{AB_{wt}} + \frac{B_T}{AB_{wt}}}{\frac{b}{AB_{wt}} + 1} - PPI\right)}{PPI \times \left(\frac{b}{AB_{wt}} + 1\right) - \frac{b}{AB_{wt}}}\right)$$

$$\Delta G = RT \times log\left(\frac{AB_{wt} \times (X - PPI) \times (Y - PPI)}{PPI \times (K+1) - K}\right) \tag{4}$$

With $K = \frac{b}{AB_{wt}}$, $X = \frac{\frac{b}{AB_{wt}} + \frac{A_T}{AB_{wt}}}{\frac{b}{AB_{wt}} + 1}$ and $Y = \frac{\frac{b}{AB_{wt}} + \frac{B_T}{AB_{wt}}}{\frac{b}{AB_{wt}} + 1}$

*Equation (4)* expresses ΔG (or, more accurately, its approximation in arbitrary units) as a function of PPI scores with the terms $AB_{wt}$, $X$, $Y$ and $K$. The smallest of the two terms $X$ and $Y$ represents the maximal PPI score, including background growth, that a variant can achieve when all the molecules of the limiting factor are in complex and assuming no change in total abundance (multiplying both side of the fraction by $AB_{wt}$ returns back to *equation 3* with $A_T$ or $B_T$ replacing AB).

It is more practical to express PPI score as a function of ΔΔG rather than ΔG:

$$\Delta\Delta G = \Delta G - \Delta G_{wt}$$

$$= RT \times log\left(\frac{\frac{AB_{wt} \times (X - PPI) \times (Y - PPI)}{PPI \times (K+1) - K}}{\frac{AB_{wt} \times (X - PPI_{wt}) \times (Y - PPI_{wt})}{PPI_{wt} \times (K+1) - K}}\right)$$

The PPI score of the wild-type is one by definition, and $AB_{wt}$ cancels out from the numerator and the denominator, leading to

$$\Delta\Delta G = \mathrm{R}T \times log\left(\frac{1}{(PPI \times (K+1) - K)} \times \frac{(X - PPI) \times (Y - PPI)}{(X - 1) \times (Y - 1)}\right) \quad (5)$$

Rearranging *Equation (5)*,

$$(X - PPI) \times (Y - PPI) - (PPI \times (K+1) - K) \times (X - 1) \times (Y - 1) \times e^{\frac{\Delta\Delta G}{\mathrm{R}T}} = 0$$

$$PPI^2 - PPI \times \left((X - 1) \times (Y - 1) \times (K+1) \times e^{\frac{\Delta\Delta G}{\mathrm{R}T}} + X + Y\right) + K \times (X - 1) \times (Y - 1) \times e^{\frac{\Delta\Delta G}{\mathrm{R}T}} + XY = 0 \quad (6)$$

*Equation (6)* is a quadratic equation with two solutions but a single relevant solution (the other root would lead to PPI scores higher than the maximum dictated by the free parameters):

$$PPI = \tfrac{1}{2} \times \left((X - 1) \times (Y - 1) \times (K+1) \times e^{\frac{\Delta\Delta G}{\mathrm{R}T}} + X + Y\right.$$

$$\left. -\sqrt{\left((X - 1) \times (Y - 1) \times (K+1) \times e^{\frac{\Delta\Delta G}{\mathrm{R}T}} + X + Y\right)^2 - 4 \times \left(K \times (X - 1) \times (Y - 1) \times e^{\frac{\Delta\Delta G}{\mathrm{R}T}} + XY\right)}\right) \quad (7)$$

According to the null expectation, the ΔΔG of a double mutant *d* is the sum of the ΔΔG of the two corresponding single mutants *s1* and *s2* (*Horovitz, 1996*), such that

$$\Delta\Delta G_d = \Delta\Delta G_{s1} + \Delta\Delta G_{s2}$$
$$= \mathrm{R}T \times log\left(\frac{(X - PPI_{s1}) \times (Y - PPI_{s1})}{(PPI_{s1} \times (K+1) - K)} \times \frac{(X - PPI_{s2}) \times (Y - PPI_{s2})}{(PPI_{s2} \times (K+1) - K)} \times \frac{1}{((X-1) \times (Y-1))^2}\right) \quad (8)$$

Replacing *Equation (8)* in *Equation (7)* leads to:

$$PPI_d = \tfrac{1}{2} \times \left(\frac{(K+1) \times (X - PPI_{s1}) \times (Y - PPI_{s1}) \times (X - PPI_{s2}) \times (Y - PPI_{s2})}{(PPI_{s1} \times (K+1) - K) \times (PPI_{s2} \times (K+1) - K) \times (X-1) \times (Y-1)} + X + Y\right.$$

$$\left. -\sqrt{\left(\frac{(K+1) \times (X - PPI_{s1}) \times (Y - PPI_{s1}) \times (X - PPI_{s2}) \times (Y - PPI_{s2})}{(PPI_{s1} \times (K+1) - K) \times (PPI_{s2} \times (K+1) - K) \times (X-1) \times (Y-1)} + X + Y\right)^2 -4 \times \left(\frac{K \times (X - PPI_{s1}) \times (Y - PPI_{s1}) \times (X - PPI_{s2}) \times (Y - PPI_{s2})}{(PPI_{s1} \times (K+1) - K) \times (PPI_{s2} \times (K+1) - K) \times (X-1) \times (Y-1)} + XY\right)}\right) \quad (9)$$

*Equation (9)* represents the thermodynamic model that predicts the PPI score of the double mutant as a function of the PPI scores of the two corresponding single mutants and has three free parameters $\frac{A_T}{AB_{wt}}$, $\frac{B_T}{AB_{wt}}$ and $\frac{b}{AB_{wt}}$.

The model was fitted to the data with an exhaustive parameter search from 1.1 to 2 with a step of 0.01 for $\frac{A_T}{AB_{wt}}$ and $\frac{B_T}{AB_{wt}}$ and from 0.3 to 1.2 with a step of 0.001 for $\frac{b}{AB_{wt}}$. These intervals were chosen based on the distribution of PPI scores, for example a background growth of > 1.2 would not make sense since it would mean that everything is background growth. As can be seen in *Figure 3—figure supplement 2A*, it is fair to assume that the optimal fit lies within these intervals. In the cases where one or both single mutants had a PPI score, respectively, below or above the bottom or top asymptotes defined by *equation (7)* and illustrated on *Figure 3D*, *equation (9)* would give infinite or aberrant (negative) values. The corresponding predicted double mutant PPI score was hence set to the minimal and maximal values, respectively, allowed by the function $((\frac{b}{AB_{wt}})/(\frac{b}{AB_{wt}}+1)$ and $\min(\frac{A_T}{AB_{wt}}, \frac{B_T}{AB_{wt}})/(\frac{b}{AB_{wt}}+1)$, respectively).

For the *trans* library alone (first part of the paper), the model was fitted on the three replicates pooled together and the parameter set leading to the lowest percentage of variance explained between the observed and predicted double mutants PPI scores was then chosen. To ensure that the model was not over-fitting, a Monte-Carlo cross-validation was performed as follow. One half of the Fos and Jun single mutants were randomly sampled and the corresponding double mutants were used as a fitting set. The double mutants corresponding to the other halves of the Fos and Jun single mutants constituted the test set, to ensure independency of the two sets. The model was then fitted with an exhaustive parameter search from 1.1 to 1.6 with a step of 0.01 for $\frac{A_T}{AB_{wt}}$ and $\frac{B_T}{AB_{wt}}$ and from 0.3 to 1 with a step of 0.01 for $\frac{b}{AB_{wt}}$. The procedure was repeated 1000 times and the median

value of the three fitted free parameters matched the values obtained from fitting the model on the whole dataset, indicating no over-fitting. The median of the proportion of variance explained for the 1000 test sets differed by only 0.001 with a standard error of $1.2 \times 10^{-4}$ across the 1000 random samples (*Figure 3—figure supplement 2B*).

## Calculation of structural genetic interaction scores

Genetic interaction scores for the three replicate of each double mutant were calculated as the difference between the observed and the predicted PPI scores and averaged across the three replicates. To test whether the genetic interaction scores were significantly different from 0, a one-sample t-test was performed. p-Values were then corrected using a permutation procedure as described above for genetic interactions calculated from the multiplicative model.

## Enrichment in strong structural genetic interactions per pair of positions, between position types and at different distance thresholds

Enrichments for cases of either strong positive or strong negative genetic interactions (absolute genetic interaction score > 0.1 and FDR < 0.2) were tested using Fisher's exact test. FDRs were controlled independently for enrichment in positive and negative genetic interactions and for each feature tested. Significant enrichments or depletions were determined at an FDR of 10%. Distance between mutated positions were measured from the crystal structure (pdb 1fos, [*Glover and Harrison, 1995*]). The smallest distance between side chain atoms was used. The robustness of these enrichments was tested by repeating this analysis using different absolute genetic interaction score cut-offs.

## *cis* library

### Experimental design

For the *cis* library, *FOS* was mutated by doped oligonucleotide synthesis instead of using NNS mutagenic primers, which would have required an iterative mutagenic process. A doping rate of 2.1% was chosen to maximize the frequency of double nucleotide mutants at 0.274, or $6.66 \times 10^{-6}$ for each of the 41,040 double mutants. Accounting for ~ 25% of non-workable, indel-containing sequences, $f_{low}$ was estimated at ~ $5 \times 10^{-6}$. Using the same rule of thumb than for the *trans* library, we targeted $10/f_{low} = 2 \times 10^6$ transformants in bacteria and yeast. Selections and competitions were performed in 200 mL in order to have a bottleneck size and a number of generations similar to the *trans* library during the competition assay (cell density of $10^7$ cells / OD/ mL * 0.05 OD * 200 mL * $f_{low} \approx 500$ cells per variant). However, the six samples (three replicates input and output) were still sequenced on a whole lane, leading to expected increase in coverage per double mutant (~30M reads per library * $f_{low} \approx 200$ reads per sample). A total of $2 \times 10^8$ template molecules per sample were targeted for the sequencing library preparation.

### Construction of the *cis* library by Gibson assembly of a doped oligonucleotide

The doped oligonucleotide was purchased from TriLink and cloned into pGD012 by Gibson assembly. The backbone fragment was prepared as for the *trans* library. The upstream *FOS* fragment and terminator + *JUN* fragment were amplified by PCR using using Kapa HiFi HotStart DNA polymerase (Kapa Biosystems, Wilmington, Massachusetts) according to the manufacturer's protocol, each in 4 × 50 µL with 1 ng of pGD012 template DNA and 15 pmol of each primer (OGD145-OGD271 and OGD138-OGD272) per 50 µL reaction and using a melting temperature of 63°C and 58°C, respectively, and an extension time of 1 min for 24 cycles. The PCR products of correct sizes were purified on agarose gel using QIAEX II (Qiagen, Netherlands) beads according to manufacturer's protocol, using 10 µL beads and eluted in 2 × 20 µL of EB buffer. The doped oligonucleotide was prepared by complementary strand synthesis using Q5 Hot Start High-Fidelity DNA Polymerase (New England Biolabs, Ipswich, Massachusetts) according to manufacturer's protocol with 12 pmol of template doped oligonucleotide and 12.5 pmol of primer OGD270 per 25 µL reaction. A temperature gradient of 12 reactions between 50°C and 72°C was first performed to determine the optimal hybdrization temperature, with an extension time of 30 s and 32 cycles to make sure all complementary strands are synthesized. Melting temperature didn't seem to affect specificity and the double

stranded DNA from the first eight reactions were each purified on a lane of an E-gel SizeSelect 2% (Invitrogen, Carlsbad, California), pooled and then concentrated on minElute column (Qiagen, Netherlands) according to manufacturer's protocol and eluted in $2 \times 10$ μL EB buffer.

The four fragments were assembled by Gibson assembly in 50 μL with 137.5 fmol of backbone fragment, 343.75 fmol (2.5X) of each of the upstream and downstream fragments and 1.375 pmol (10X) of the double stranded doped oligonucleotide. After a 24 hr incubation at 50℃, the reaction was dialyzed for 1 hr on nitrocellulose membranes with 0.025 μm pores (Merck Millipore, Burlington, Massachusetts). A single batch of transformation was realized as described for the *trans* library and pooled for 25 min recovery (to account for the time it takes to do all eight electroporations and ensure cells from the first electroporation didn't divide yet). After this recovery period, 2 μL were diluted 100 X in the same medium and 100 μL were plated on pre-warmed LB + Amp Petri dishes and incubated for ~16 hr at 37℃. The remaining ~ 16 mL of cells were added to 400 mL of pre-warmed LB + 400 μg/mL in 1L flasks and incubated for ~16 hr at 37℃ under constant agitation at 200 rpm. This resulted in a total of ~$1.75 \times 10^6$ transformants. The cells were harvested and the plasmids were extracted as described for the *trans* library.

## Large-scale yeast transformation

The transformation protocol was further optimized between the transformation of the *trans* and *cis* libraries and scaled down for the latter since it is of lower complexity. First, a glycerol stock of the BY4742 strain was streaked to single colonies on YPD Petri dishes and incubated at 30℃ for 2 days. For each replicate, a whole colony was re-suspended in 20 YPDA for the *cis* and *trans* libraries, respectively, and grown overnight to saturation at 30℃ under constant agitation at 200 rpm. The next day, three flasks of 175 mL of pre-warmed YPDA were inoculated with each of the pre-culture at a starting $OD_{600nm}$ of 0.3 and incubated at 30℃ under constant agitation at 200 rpm. After 4 hr of incubation, the cells were harvested by centrifuging for 5 min at 3000 *g* in an Avanti J-26 XP Centrifuge using a JLA 8.1000 rotor (Beckman-Coulter, Brea, California). Cells were washed once with 50 mL sterile distilled water, re-suspended in 10 mL SORB, transferred to 50 mL Falcon tubes and completed to 25 mL with SORB. The cells were harvested by centrifugation 5 min at 3000 *g* in an Eppendorf Centrifuge 5810 R (Eppendorf, Germany), re-suspended in 8.6 mL SORB and incubated at room temperature on a wheel for 30 min. Meanwhile, 1.1 mL of 10 mg/mL salmon sperm DNA (Agilent Technologies, Santa Clara, California) were thawed, boiled for 5 min in a heating block and cooled on ice for 2 min. When the cells were ready, 3.5 μg of plasmid DNA library, 175 μL salmon sperm DNA and 35 mL of plate mixture were added to each replicate and mixed by inverting the tubes six times. The Falcon tubes were incubated at room temperature for 30 min on a SSM4 mini seesaw rocker (Stuart Equipment, United Kingdom) at 30 oscillations/min. Then, 3.5 mL of DMSO were added to each tube and mixed in by inverting the tubes six times. The heat-shock procedure was identical as for the *trans* library. The cells from each tube were re-suspended in 10 mL of pre-warmed recovery medium and added to 350 mL of pre-warmed recovery medium in 1 L flasks. Recovery medium were incubated for 1 hr at 30℃ under constant agitation at 200 rpm. The cells were then harvested by centrifugation at 3000 *g* for 5 min in a Avanti J-26 XP Centrifuge using a JLA 8.1000 rotor (Beckman-Coulter, Brea, Californi), washed once with 25 mL of SC –ura, respectively, and added to 350 mL of SC –ura in 2 L flasks. After homogenization by stirring, 10 μL were platted on SC –ura Petri dishes and incubated for ~48 hr at 30℃ to measure transformation efficiency. The liquid cultures were grown to saturation for ~48 hr at 30℃ under constant agitation at 200 rpm. This transformation resulted in $1.17 \times 10^7$, $9.6 \times 10^6$ and $1.09 \times 10^7$ transformants for replicates 1, 2 and 3, respectively.

## Competition assay

The competition was performed in the same way as for the *trans* library, except for the following minor changes. The post-transformation selection cycle and the two cycles of competition were performed in 200 mL in 500 mL flasks (the *cis* library is of lower complexity hence smaller volumes are sufficient). The first cycles of competition were grown to an $OD_{600nm}$ of 1.585, 1.615 and 1.65 for the three replicates, respectively. The second cycle of competition was inoculated at an $OD_{600nm}$ of 0.1 and the three replicates were harvested at an $OD_{600nm}$ of 1.475, 1.59 and 1.49, respectively. Finally,

150 mL of the post-transformation selection and competition cultures were harvested at the end of the cycles for DNA extraction.

## DNA extraction

The *cis* library was processed in the same way as the *trans* library, except for the following minor changes. Because the DNA was extracted from 150 mL instead of 1 L cultures, all the volumes were scaled down by 0.15 X. Cell lysis and the two phenolic extractions were performed the same way. In between the processing of the two different libraries, the protocol was simplified and improved to get rid of the residual SDA and impurities more efficiently. DNA was first precipitated in the same way as previously but using 0.1 volumes of 3 M NaOAc instead of 2 M NaCl. After pelleting and drying overnight, the pellets were re-suspended in 900 µL TE buffer. As for the *trans* samples, the pellets didn't re-suspend well and a lot of insoluble material remained. The samples were treated with 7.5 µL RNAseA 10 mg/ml and incubated 30 min at 37°C. DNA was then purified using 75 µL QIAEX II beads (Qiagen, Netherlands) per sample following the manufacturer's protocol. DNA was eluted in 2 × 187.5 75 µL after 5-min incubation at 60°C.

## Sequencing library preparation

The preparation of the *cis* library for sequencing was performed the same way as for the *trans* library, except for the following minor changes. First, only the Fos region was amplified, from directly upstream of the first mutated codon to directly downstream of the last mutated codon. For the first round of PCR, only 4 × 50 µL reactions per sample were performed, with $50 \times 10^6$ molecules of plasmid in each reaction and primers OGD277 and OGD278 at a melting temperature of 56°C and an extension time of 30 s. The four PCRs from the same sample were pooled and purified using 12 µL (6 µg) QIAEX II beads (Qiagen, Netherlands). The eluted DNA was then split into two PCR reactions per sample, which were performed in the same way as for the *trans* library. The two PCRs of each sample were then pooled but not further concentrated. Relative DNA concentrations were measured as for the *trans* library but directly from the PCR reactions. The six samples were then pooled at equimolar ratio and the correct band was purified from 100 µL on a 1% agarose gel using 4 µL of QIAEX II beads (Qiagen, Netherlands). The pooled library was sequenced in a single lane of an Illumina HiSeq2500 with 125 bp paired-end reads at the EMBL Genomics Core Facilities in Heidelberg, Germany.

## Sequencing data analysis and filtering

A total of 262,393,254 read pairs were obtained from the lane, with ~ 12% adapter sequences. However, these were not filtered at this step. Paired-end reads, which were overlapping over the full length of the variable region, were assembled using *PEAR* version 0.9.6 (*Zhang et al., 2014*) with a p-value threshold for correct assembly of 0.05, a maximum and minimum fragment length of 150 nt, and a minimum overlap size of 100 nt. These parameters force the assembly of sequences of the unique expected length. An average of 30% and 20% of the paired end reads from the three input and output libraries, respectively, were not assembled and filtered out. These included the adapter sequences and indels (resulting from the <100% coupling efficiency inherent to DNA synthesis). Additionally, assembled reads with Ns in any part of the sequence were filtered out. These filtering steps resulted in between 3.0 and $3.7 \times 10^7$ usable reads per replicate input and output.

## PPI score calculation

PPI scores were calculated in the same way as for the *trans* library. However, all amino acid variants with more than two substitutions were filtered out along with amino acid substitution that require more than one nucleotide substitution from the WT sequence. This was required because double amino acids mutants for which one or both of the two mutated codons were more than one nucleotide away from the WT codon had a lower sequencing coverage and were hence less reliable. However, frequencies were calculated accounting for the total number of different UMIs of the corresponding replicate, that is accounting for all the other variants present in the sequenced population including those with more than two mutations. As for the *trans* library, variants with less than 10 UMIs in any of the three input replicates or without any UMI in any of the three output replicates

were filtered out at this step. The WT and variants containing stop codons were also excluded from downstream analyses.

### Restricting the *trans* library for direct comparison to the *cis* library

For a fair comparison of mutation effects and genetic interaction patterns between the two libraries, all analyses for the *trans* library were repeated after filtering out all variants for which the amino acid substitutions require more than one nucleotide change from the WT amino acid.

### Correction for background growth

The mode of detrimental variants was adjusted in the same way as for *trans* library to correct for batch effects in background growth. The two libraries, that is the *cis* and the restricted *trans*, were adjusted to the same value, that is the maximal value of the mode across the three replicates of the two libraries.

### Classification of mutant effects

Classification was performed using the same cut-offs as for the non-restricted *trans* library. However, FDR was re-computed for the restricted *trans* library to take into account the ~8 x decrease in the number of tests.

### Matching of the single mutant distributions between the *cis* and *trans* libraries

The differences in double mutants effects and pattern of genetic interactions in the *cis* and *trans* libraries can be the result of i) biases in double mutant compositions of the two libraries even after restricting the *trans* library to amino acid substitutions reachable through single nucleotide changes; ii) differences in single mutant effect distributions, that is combining two Fos single mutants has a different multiplicative and thermodynamic expectation than combining one Fos and one Jun; and iii) differences in structural genetic interactions in *cis* and *trans*. To control for the two effects, the two libraries were sub-sampled as follows to match the effects of their single mutant distributions. For each of 1000 sub-samplings, the two libraries were each binned in two dimensions according to the PPI scores of their constituting single mutants. Each dimension was binned into 20 evenly spaced bins from the lowest to the highest single mutants PPI score measured in the two libraries (0.477 and 1.082, respectively). For the *cis* library, the two single mutants constituting each double mutant were assigned to one or the other dimension randomly in each sub-sampling. For each two-way bin, the double mutants belonging to the library that was the most represented in the bin were randomly sub-sampled to match the number of double mutants belonging to the same bin in the other library. This resulted in identical distributions of single mutants PPI scores according to these binning breaks.

### Comparisons of proportions of double mutants of different effect classes between the *cis* and *trans* libraries

Double mutants from each of the 1000 sub-sampled libraries of matched single mutant effects were classified using the same class definitions as above. The proportions of each class in each library averaged over the 1000 matched sub-samples were reported. The p-value corresponds to the proportion of matched sub-samples where the differences between the two libraries were in the opposite direction than the averages.

### Thermodynamic model

The same thermodynamic model as above (*Equation 9*) was fitted. All fitting procedures were performed after pooling the three replicates in the same model. To test whether the same model with the same parameter would fit both the *cis* and *trans* libraries, the model was first fitted on both libraries independently. These libraries were the original ones, that is before matching single mutant distributions (but the restricted *trans* library). The two fitting procedures converged to similar parameters that had an almost identical predictive power when tested on the other library than the one it was fitted on. To test whether the differences in the distribution of single mutant effects between

the two libraries would affect the fitting, this was repeated on the 1000 matched, sub-sampled libraries. This confirmed that the model was robust to and independent of these differences. The parameters and genetic interaction scores used in subsequent analyses were thus the ones from the model fitted on the two libraries pooled together.

Genetic interaction scores were calculated as the difference between the observed double mutants PPI scores and the ones predicted by the thermodynamic model. The significance of genetic interactions was tested using a one-sample t-test. The FDR at each p-value threshold was then calculated independently for the two libraries to account for the differences in the p-value distributions that might be partly the consequences of different levels of noise or technical artifacts in the two libraries. This was performed as explained above using 10,000 permutations.

The proportion of the non-random variance in the double mutants PPI scores explained by the model for each library was calculated first for the original libraries. The proportion of total non-random variance was calculated as the average of the squared Pearson's correlation coefficients between the double mutant PPI scores of each pair of the three replicates of a given library. The proportion of non-random variance explained by the model was then calculated as the percentage ovariance explained by the model divided by the proportion of total non-random variance.

To test whether this proportion of non-random variance explained by the model is independent of the differences in single mutant effects, this was repeated on 1000 libraries sub-sampled to match their distribution of single mutants effects. The proportion of total non-random variance was recalculated in each sub-sample because they might differ since the double mutant compositions differ. The model was not re-fitted on each sub-sample since it was demonstrated above that this does not vary significantly. The proportion of non-random variance explained by the model was then recalculated and averaged across the 1000 matched sub-samples.

## Comparisons of the proportion of structural genetic interaction in *cis* and *trans*

To compare the proportion of structural positive and negative genetic interactions in *cis* and *trans*, the same p-value threshold was chosen in the two libraries instead of the same FDR because identical variants with identical effect sizes and identical levels of noise, that is variants that would have identical p-values after the t-test, could otherwise be significant in one but not in the other library due to different FDR adjustments. The average proportion of true cases of genetic interactions was then estimated in each library as the total number of significant cases of either positive or negative genetic interactions multiplied by one minus the average FDR calculated by permutation at any given p-value threshold.

To ensure that the higher prevalence of both positive and negative genetic interactions is not the consequence of different single mutant effect sizes in the two libraries, this analyses was repeated on the 1000 libraries randomly sub-sampled to match their distributions of single mutants effects. In each sub-sample, FDRs were re-computed using 10,000 permutations as explained above. The proportion of both positive and negative genetic interactions was then computed in each sub-sample using the FDR averaged over the 1000 randomizations.

## Enrichment in strong structural genetic interactions per pair of positions, between position types and at different distance thresholds

Enrichments for cases of either strong positive or strong negative genetic interactions (absolute genetic interaction score > 0.1 and FDR < 0.2) were tested by Fisher's exact test as described above for the non-restricted *trans* library alone. For this analysis, different p-value thresholds leading to the same FDR in the two libraries were used to minimize the differences in noise levels between the two libraries.

## Acknowledgements

This work was supported by a European Research Council (ERC) Consolidator grant (616434), the Spanish Ministry of Economy and Competitiveness (BFU2011-26206 and 'Centro de Excelencia Severo Ochoa' SEV-2012–0208), the AXA Research Fund, the Bettencourt Schueller Foundation, Agencia de Gestio d'Ajuts Universitaris i de Recerca (AGAUR, SGR-831), the EMBL-CRG Systems Biology

Program, and the CERCA Program/Generalitat de Catalunya. GD is a non-stipendiary EMBO Fellow and a Marie-Curie Fellow under grant agreement 608959.

## Additional information

### Funding

| Funder | Grant reference number | Author |
| --- | --- | --- |
| European Research Council | 616434 | Ben Lehner |
| AXA Research Fund | Professorship | Ben Lehner |
| Ministerio de Economía y Competitividad | BFU2011-26206 and SEV-2012-0208 | Ben Lehner |
| Agència de Gestió d'Ajuts Universitaris i de Recerca | SGR-831 | Ben Lehner |
| Fondation Bettencourt Schueller | Bettencourt Prize | Ben Lehner |
| European Commission | Marrie-Currie co-fund fellowship 608959 | Ben Lehner |
| EMBL-CRG | Systems Biology Program | Ben Lehner |
| Generalitat de Catalunya | CERCA Program | Ben Lehner |

The funders had no role in study design, data collection and interpretation, or the decision to submit the work for publication.

### Author contributions

Guillaume Diss, Conceptualization, Data curation, Software, Formal analysis, Funding acquisition, Validation, Investigation, Visualization, Methodology, Writing—original draft, Writing—review and editing; Ben Lehner, Conceptualization, Supervision, Funding acquisition, Writing—original draft, Writing—review and editing

### Author ORCIDs

Guillaume Diss (iD) https://orcid.org/0000-0001-9153-4523
Ben Lehner (iD) https://orcid.org/0000-0002-8817-1124

### Decision letter and Author response

Decision letter https://doi.org/10.7554/eLife.32472.033
Author response https://doi.org/10.7554/eLife.32472.034

## Additional files

### Supplementary files

• Supplementary file 1. Data from the full *trans* library.
DOI: https://doi.org/10.7554/eLife.32472.024

• Supplementary file 2. Variance explained by physico-chemical amino acid features.
DOI: https://doi.org/10.7554/eLife.32472.025

• Supplementary file 3. Data from the restricted *trans* library.
DOI: https://doi.org/10.7554/eLife.32472.026

• Supplementary file 4. Data from the *cis* library.
DOI: https://doi.org/10.7554/eLife.32472.027

• Supplementary file 5. Primer sequences
DOI: https://doi.org/10.7554/eLife.32472.028

• Transparent reporting form
DOI: https://doi.org/10.7554/eLife.32472.029

## Major datasets

The following dataset was generated:

| Author(s) | Year | Dataset title | Dataset URL | Database, license, and accessibility information |
|---|---|---|---|---|
| Diss G, Lehner B | 2017 | The genetic landscape of a physical interaction | http://www.ncbi.nlm.nih.gov/geo/query/acc.cgi?acc=GSE102901 | Publicly available at the NCBI Gene Expression Omnibus (accession no: GSE102901). |

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
