## [Decision Letter]

Thank you for submitting your article "The genetic landscape of a physical interaction" for consideration by *eLife*. Your article has been reviewed by three peer reviewers, and the evaluation has been overseen by Naama Barkai as the Senior and Reviewing Editor. The following individual involved in review of your submission has agreed to reveal their identity: Jesse D Bloom (Reviewer #3).

The reviewers have discussed the reviews with one another and the Reviewing Editor has drafted this decision to help you prepare a revised submission.

As you will see below, the reviewers greatly appreciated the approach, but were worried about technical details. Please address all of the concerns that were raised. Most important are the concerns regarding sequencing errors, controls, and the use of the same thermodynamic model to two different aspects of the problem.

Here are the comments that I drew out of the reviews that I think should be noted. I also agree with many of the other listed points, but these are the ones that most stand out to me:

1) I agree with reviewer #1 that the authors should justify the cutoff of >10 reads. I don't really have a problem with this cutoff, but some justification would be nice. I strongly agree with #1 that the filtering to require >0 counts in selected library doesn't make any sense. A deleterious mutation would be expected to go to zero. So as #1 says, the authors should either remove this filter or come up with a really good reason why it is justified.

2) Reviewers 2 and 3 both think that it doesn't make sense to apply the same thermodynamic model to cis interactions (which presumably relates to protein stability) and trans interactions (which presumably related to binding affinity).

3) Several noted that the availability of the computer code is clearly inadequate.

4) Several noted any clear description of how they dealt with sequencing error, and the lack of controls to quantify this.

5) I agree with reviewer #2's point that the correlations should be reported in terms of the actual quantity on which inferences are based, which is PPI.

6) I agree with reviewers #1 and #2 that they need to justify the 0.64 / 0.96 / 1.04 cutoffs.

*Reviewer #1:*

The manuscript "The genetic landscape of a physical interaction by Lehner and Diss presents methods for detecting genetic interactions between combinations of point mutations in two genes encoding physically interacting proteins. The combination of Deep Mutational Scanning (DMS) with a Protein Fragment Complementation assay is innovative. They first define interactions in the usual multiplicative way, and then (most interestingly) present a thermodynamic model which explains a large fraction of the observed interactions (stemming from the idea that function depends non-linearly on folding energy). They then essentially redefine genetic interaction to look for combinatorial effects that are not explained by this null model. The combinatorial library construction method that directly establishes a genetic link between the interacting variants is an elegant experimental design. The paper is well written and provides novel insights into physical underpinnings of genetic interactions.

Despite overall enthusiasm, there are some points that should be addressed:

- The sequencing results were filtered to only consider clones with > 10 reads in the input library. How was this cutoff chosen? A plot of #reads vs. variance of technical replicates might be helpful to show demonstrate the effect of this cutoff. The rule of thumb for counts data is that the standard deviation is the square root of the count, so that a 10-count measurement will have a CV of at least 33%. Is this acceptable?

- The results were also filtered to only consider clones with >0 counts in the selected library. This was a bad idea. The mutational combinations that cause interactions to disappear altogether might correspond to some of the strongest and most interesting negative genetic interactions. Fixing this mistake may or may not have much impact on the results, but this analysis error should be corrected.

- R^2^ can be a misleading indicator of data quality. In measuring agreement between thermodynamic model and results, it would be useful to examine RMSD for held-out data not used in fitting the model.

- The expression "positive (compensatory)" is used, as though these were synonyms. In genetic interaction studies with sufficient resolution to capture different subtypes of positive interactions (see St Onge et al., Nat Genetics 2007), the bulk of positive interactions have been "masking" or "diminishing returns" interactions, where the first mutation disrupted a process, and the second disruptive mutation could do no further harm because the damage was already done. To be suppressive or compensatory means that not only must the combination of mutations be above multiplicative (or thermodynamic?) expectation, but the combination must cause less harm to the phenotypes than the most harmful of the two mutations. Adding specific separate analyses of masking and suppressive/compensatory positive interactions could greatly improve the impact of this study, and this would not be hard to do.

- It is said that two mutations within Fos are more likely to increase the strength of the PPI than one mutation in Fos combined with a second mutation in Jun. To complete the argument, the authors should also examine whether two mutations within Jun are more likely to increase the strength of the PPI than one mutation in Jun combined with a second mutation in Fos.

- In Figure 4E, the last example (bottom right panel) does not seem to make sense biochemically. While the WT and double mutant both can be expected to have a hydrophobic interaction, the individual single mutants would also be expected to be able to fulfill that role.

- "Sequencing data was filtered using homemade Perl scripts" is hardly a reproducible description, especially given that there is no statement about code availability. The code written for this study (e.g. perl scripts for sequence analysis, R scripts for statistical analyses) would ideally be publicly posted (e.g., at Github) and not simply "available on request".

- Some thresholds used for classifying mutant effects seem to come from nowhere. While the 1.04 threshold is justified in terms of its control of FDR at 0.05, no similar justification is provided for 0.64 or 0.96 thresholds.

- Subsection “Comparisons of the proportion of structural epistasis in *cis* and *trans*”. It is not at all clear that using the same P-value threshold allows apples-apples comparison between different libraries of different sizes. Would be better to computationally down-sample the larger of the two libraries.

*Reviewer #2:*

This paper uses deep mutational scanning (DMS) on two proteins to study the genetic determinants of binding across a protein-protein interaction interface; further, it interprets these observations in a structural context. It is the first study to analyze large-scale epistasis across an interface, so it will be of considerable interest to those interested in sequence-structure-function relationships and the determinants of PPIs. I find the conceptual framework and design to be solid. The experiments appear to have been well executed and the data have been carefully analyzed, so I think the paper is a good candidate for *eLife*.

A few comments follow. I am mostly concerned that better attention to analysis of error and reproducibility is necessary. I expect that it should be possible to address my comments with a handful of additional statistical analyses and more precise description in the text.

1) Stochastic error in inferring growth rate from frequencies has not been adequate attention. The authors use all sequences with >10 input reads and >0 post-selection reads. Estimates of PPI scores (and, in turn, of epistasis) for genotypes with low read numbers will have considerable stochastic error associated with them, particularly if that error is propagated into products or ratios (as they are for epistatic effects). Although reproducibility may be relatively high overall, the authors should incorporate uncertainty caused by stochastic error into their classifications and quantifications of mutational and epistatic effects.

2) The authors claim that reproducibility is high, and refer to a high rank-order correlation coefficient among replicates. The paper's major inferences, however, are not based on rank-order but on PPI score and statistics derived from it. What matters, then, is the reproducibility of the PPI score. The authors should instead report R^2^ for PPI among reps and between DMS and small-scale experiment. If the R^2^ is low, discussion of the implications for the authors' claims will be required.

3) Because the library was created using doped degenerate codons rather than targeted synthesis, there are a great variety of genotypes in the library with different numbers of mutations. Paired-end sequencing is used (rather than bar-coding) to identify genotypes. Sequencing error may lead to mis-identification of genotypes, and this error may be biased towards genotypes with certain numbers of mutations. This concern does not appear to have been addressed. Reporting the degree of sequencing error and showing that it does not strongly affect the claims is necessary.

4) I find the PPI score unintuitive. In theory, it might range asymmetrically around the value of 1 (wildtype) from 0 for a totally deleterious genotype to infinity. This alone bothers me (and could be addressed with a log-scale relative to wildtype). The assay itself puts practical limits on how much a mutant genotype might possibly increase the PPI, because there must be saturation point at which occupancy of the complex, given the dose of methotrexate in the assay, no longer improves fitness. The PPI numbers are therefore somewhat hard to interpret. I would prefer a transform that expresses the PPI on a log scale relative to wild type, rescaled so the intervals between the minimum and wild-type and that between wild-type and maximum are the same. I don't view the authors' method as fatally flawed, and I have no reason to think the conclusions would change if my method were used. I'd like the authors to consider the possibility that such an approach might make the numbers in the paper easier for the reader to interpret.

5) I like the authors' efforts to quantitatively distinguish specific structural epistatic interactions from general thermodynamic epistasis caused by the nonlinear relationship between the proteins' affinity for each other and occupancy of the complex. Although the model has a nice theoretical basis and accounts for a substantial amount of epistasis, there may be additional causes of nonspecific epistasis remaining in the system – for example, if there are nonlinearities in the relationship between occupancy of the jun-fos complex and growth rate. I would therefore like to see the authors explore using Sailer and Harms' Genetics 2017 method, which identifies general epistasis in a way that is more agnostic to the specific biochemical causes or quantitative relationships, to determine if this is the case. If general epistasis can be identified using Sailer's approach even after the thermodynamic model is applied, this would be important, because it would further reduce (and probably sharpen) the cases of specific structural epistasis, and it would be an interesting finding, as well.

6) I have some concern that the thermodynamic correction may affect *cis* and *trans* interactions differently. Stated a little too simply, the model is built around the expectation that mutations in trans that additively affect the energy of complex formation will have an exponential effect on occupancy of the complex and therefore on PPI; epistasis not accounted for by this relationship represent specific structural epistasis. It is not apparent to me that the same model appropriately corrects for general thermodynamic nonlinearity among cis- acting mutations. Suppose that two mutations within a protein independently affect the dG of the native structure; they will have an exponential effect on occupancy of the native structure. It is not obvious to me that the model will account for such nonlinear effects on the effective concentration of each molecule. If so, then pairs of *cis*-mutations that in fact act additively on the protein might be classified as structural specific interactions even after the thermodynamic model is applied. This could contribute to the apparent excess of classified epistasis in cis compared to in trans after removal of thermodynamic epistasis (Figure 5). If I am misinterpreting the model, it would be helpful for the authors to better explain in the main text what the model does and doesn't account for.

7) The authors classify the magnitude of PPI effects as strongly deleterious, weakly deleterious, neutral, and beneficial based on thresholds of 0.64, 0.96, and 1.04. The authors should provide a justification these numbers, which are presented as if they are arbitrary. (They do resemble some relevant numbers from the normal distribution, but the authors should make this explicit and justify their use.)

8) The interface of jun and fos appears as if it might be isologous, such that a mutation in one partner would be expected to have precisely the same structural and functional effect as the same mutation in the other (although this prediction would be compromised by divergence between the paralogs over time). The authors might discuss this, particularly as an explanation for the correlation between the effects of a mutation in one partner and the same mutation in the other. Isology would explain correlation; divergence would explain imperfect correlation.

9) Some bZip proteins, including fos and jun family members, can homodimerize. In the current assay, such events would interfere with growth, assuming that homodimers would compete with the heterodimer for the relevant subunits. Mutations that specifically affect the affinity of homodimerization – including those at the interface – would thus affect PPI, but not through *trans*-interactions. The authors should consider this possibility and note how it might contribute to their observations.

10) The authors state that combining two mutations that positively affect growth rate leads to positive epistasis and cite Figure 3C. I believe that should be apparent as pies colored mostly green in the upper-right quadrant of the array. I don't see that in the data at all. That quadrant looks mostly yellow, suggesting little epistasis. What I believe is apparent in Figure 3C is that positive epistasis is concentrated in the rows where a weak beneficial mutation plus a deleterious mutation yield an outcome that is less deleterious than expected; this effect goes away when the thermodynamic model is applied (compare 3F). This is worth noting and, I hope, providing an explanation for.

*Reviewer #3:*

This study uses deep mutational scanning to assess the effects of essentially all single amino-acid mutations to the JUN and FOS proteins. It then uses an interesting strategy to analyze the effects of combinations of mutations to the two.

This is the first deep mutational scanning study to look at both sides of a protein-protein interface. In general, the experiments appear to be well done with adequate replicates – at least I could identify no obvious technical flaws. They also have a nice and detailed Materials and methods section.

The analysis in terms of thermodynamic effects was quite nice, and I like the result about a mix of thermodynamic and structural epistasis.

In general, unless the other reviewers identify significant problems that I have overlooked, I support publication of this paper.

* The only thing that I see lacking experimentally is deep sequencing of the wildtype sequences to estimate the error rate. Right now there is no such estimate of how many of the observed mutations are actually due to library preparation of sequencing errors. This could easily be determined by sequencing wild type.

* The computer code used for the data analysis should be included as a supplementary file and/or a GitHub repository (*eLife* now allows this).

* In various places (first in the subsection “Determinants of single mutant outcome”), the authors refer to the "identical substitutions" in both proteins. But they never clearly explain what this means. I'm assuming that the proteins are homologous, and they mean the homologous positions? This needs to be described much better.

* Unless I am misunderstanding, the thermodynamic model is in terms of mutational effects on the protein-protein binding. This makes sense for analyzing the trans mutational effects. But for analyzing the *cis* mutational effects, wouldn't we expect to instead be concerned with ddG values for the stabilities of the individual proteins?

---

## [Author Response]

Here are the comments that I drew out of the reviews that I think should be noted. I also agree with many of the other listed points, but these are the ones that most stand out to me:1) I agree with reviewer #1 that the authors should justify the cutoff of >10 reads. I don't really have a problem with this cutoff, but some justification would be nice. I strongly agree with #1 that the filtering to require >0 counts in selected library doesn't make any sense. A deleterious mutation would be expected to go to zero. So as #1 says, the authors should either remove this filter or come up with a really good reason why it is justified.

We used these read cut-offs because the low read counts lead to a lower boundary for PPI scores when the output reads equal one. Output read counts of 0 thus lead to a large uncertainty in PPI score. With 10 reads or below in the input, the PPI score can never be lower than ~0.5 (when output read count is 1), which biases those variants with low input read counts. For more details, please refer to the answers to reviewers’ comments.

2) Reviewers 2 and 3 both think that it doesn't make sense to apply the same thermodynamic model to cis interactions (which presumably relates to protein stability) and trans interactions (which presumably related to binding affinity).

Most leucine zippers, including *FOS* and *JUN*, have a coupled folding-binding equilibrium. The system can thus be modeled as a two-state equilibrium with a single ΔΔG that fits both the *cis* and *trans* double mutants.

3) Several noted that the availability of the computer code is clearly inadequate.

We now uploaded the code on GitHub at https://github.com/gdiss/Diss_et_al_*eLife*_2018 and added the link at the beginning of the Materials and methods section.

4) Several noted any clear description of how they dealt with sequencing error, and the lack of controls to quantify this.

We now estimate sequencing error and show that more stringent quality filters only affect PPI scores marginally.

5) I agree with reviewer #2's point that the correlations should be reported in terms of the actual quantity on which inferences are based, which is PPI.

We now report Pearson’s correlation coefficient instead of Spearman’s where appropriate.

6) I agree with reviewers #1 and #2 that they need to justify the 0.64 / 0.96 / 1.04 cutoffs.

We show in Figure 2—figure supplement 1B-C and Figure 5—figure supplement 2B-E that our results are robust to the choice of the cutoffs within ranges relevant to each mutant class according to the distribution of mutant effects.

Reviewer #1:[…] Despite overall enthusiasm, there are some points that should be addressed:- The sequencing results were filtered to only consider clones with > 10 reads in the input library. How was this cutoff chosen? A plot of #reads vs. variance of technical replicates might be helpful to show demonstrate the effect of this cutoff. The rule of thumb for counts data is that the standard deviation is the square root of the count, so that a 10-count measurement will have a CV of at least 33%. Is this acceptable?

We added panel B in Figure 1—figure supplement 1 and a paragraph in the Materials and methods (section”Sequencing data analysis and filtering”) to explain why we chose both the 10 input reads and 0 output read cut-offs:

“Variants with zero UMI in any of the three outputs were filtered out. Even though variants with lethal mutations could be expected to go to 0, they cannot go further down while their actual frequency in the population is still decreasing (e.g. they could still not be detected even if sequencing 10X or 100X deeper). […] We thus filtered out variants with less than 10 UMIs in any of the three input replicates”.

We now also report that our results regarding the enrichment of epistatic interactions between proximal and interfacial residues hold when increasing this threshold to 100 (Figure 4—figure supplement 3C and Figure 5—figure supplement 5)

- The results were also filtered to only consider clones with >0 counts in the selected library. This was a bad idea. The mutational combinations that cause interactions to disappear altogether might correspond to some of the strongest and most interesting negative genetic interactions. Fixing this mistake may or may not have much impact on the results, but this analysis error should be corrected.

Please refer to the answer to the previous comment.

- R^2^ can be a misleading indicator of data quality. In measuring agreement between thermodynamic model and results, it would be useful to examine RMSD for held-out data not used in fitting the model.

We agree and apologize for using the wrong term. What we report is actually the percentage of variance explained as calculated by 1- RSS/TSS, where RSS and TSS are the residual and total sum of squares, respectively. While it is also sometimes called R^2^, we understand it can be confusing as it doesn’t not equate to the actual square of the Pearson correlation coefficient. We hence replaced this term by “% var. explained” where appropriate.

We also now report the RMSD as we agree that it is relevant. As explained in the method, we use a Monte-Carlo cross-validation to fit the thermodynamic model. Briefly, double mutants are split in two independent sets representing each 25% of the data. The identity of the mutants compositing these two sets do not overlap, ensuring their independence (i.e. double mutants corresponding to 50% of the FOS and JUN single mutants, chosen randomly, were attributed to the training set and the double mutants corresponding to the other 50% of FOS and JUN single mutants were attributed to the test set). The thermodynamic model was then fitted on the training set and evaluated on the test set. This was repeated 1,000 times and the median of each of the three parameters corresponded exactly to when fitting the data on the whole dataset. The median% of variance explained was also within 0.001 of that when the model was fitted on the whole dataset.

We report in Figure 3E the% of variance explained and the RMSD for the whole dataset and in Figure 3—figure supplement 2B the median of these two values across the 1,000 Monte-Carlo cross-validations.

- The expression "positive (compensatory)" is used, as though these were synonyms. In genetic interaction studies with sufficient resolution to capture different subtypes of positive interactions (see St Onge et al., Nat Genetics 2007), the bulk of positive interactions have been "masking" or "diminishing returns" interactions, where the first mutation disrupted a process, and the second disruptive mutation could do no further harm because the damage was already done. To be suppressive or compensatory means that not only must the combination of mutations be above multiplicative (or thermodynamic?) expectation, but the combination must cause less harm to the phenotypes than the most harmful of the two mutations. Adding specific separate analyses of masking and suppressive/compensatory positive interactions could greatly improve the impact of this study, and this would not be hard to do.

We agree that all positive interactions are not necessarily compensatory and apologize for the imprecision. We have removed this term. As for the separate analyses of masking vs. compensatory interactions, we believe these go beyond the scope of the present study.

- It is said that two mutations within Fos are more likely to increase the strength of the PPI than one mutation in Fos combined with a second mutation in Jun. To complete the argument, the authors should also examine whether two mutations within Jun are more likely to increase the strength of the PPI than one mutation in Jun combined with a second mutation in Fos.

We agree that this comparison would be interesting. However, we make this statement based on the actual comparison between the *trans* and *cis FOS* double mutant data. We did not assay the effects of *cis* double mutants in *JUN* and can thus unfortunately not do the same comparison.

- In Figure 4E, the last example (bottom right panel) does not seem to make sense biochemically. While the WT and double mutant both can be expected to have a hydrophobic interaction, the individual single mutants would also be expected to be able to fulfill that role.

This example can indeed be counter-intuitive. However, a speculative explanation could be the following. As can be seen on the figure, the two facing Leucines side chains extend outward of the interface and in opposite directions, so that the bulky, branched end of the side chain do not clash with each other. Valine has one less carbon atom in the side chain than leucine but the same branched carbon at the end of the side chain. A mutation from Leucine to Valine in FOS might therefore bring the branched end of the side chain closer to the interface, which could clash with the facing Leucine in JUN. We indeed see a PPI score of 0.71 for this single mutant. On the contrary, the less bulky Alanine mutation in JUN appears to be near neutral (PPI score = 0.95). At the same time, it makes room so that the Leu-to-Val mutation in FOS does not clash anymore, such that the double mutant is more stable than the Val single mutant (PPI score = 0.91).

- "Sequencing data was filtered using homemade Perl scripts" is hardly a reproducible description, especially given that there is no statement about code availability. The code written for this study (e.g. perl scripts for sequence analysis, R scripts for statistical analyses) would ideally be publicly posted (e.g., at Github) and not simply "available on request".

We now uploaded the code on GitHub at https://github.com/gdiss/Diss_et_al_*eLife*_2018 and added the link at the beginning of the Materials and methods section.

- Some thresholds used for classifying mutant effects seem to come from nowhere. While the 1.04 threshold is justified in terms of its control of FDR at.0.05, no similar justification is provided for 0.64 or 0.96 thresholds.

We agree that these thresholds are arbitrary. However we show in Figure 2—figure supplement 1B-C and Figure 5—figure supplement 2B-E that our conclusions hold when modifying the thresholds within relevant ranges for the class they delineate based on the distribution of PPI scores.

- Subsection “Comparisons of the proportion of structural epistasis in cis and trans”. It is not at all clear that using the same P-value threshold allows apples-apples comparison between different libraries of different sizes. Would be better to computationally down-sample the larger of the two libraries.

We understand that this can be unintuitive (we actually had to think about how to make these comparisons for quite a while), which is why we also reported comparisons after 1000 random down-samplings of the larger library (Figure 5—figure supplement 2, 3C and 9).

However, we would like to keep the comparison at the same p-value threshold because we believe it is valid, as explained in the Materials and methods. Indeed, two identical mutations in the two libraries will have the exact same effect on the PPI and therefore ought to be identically classified as significantly detrimental or beneficial. If the technical noise is the same in the two libraries, these two variants should have the same measured mean effect and the same standard error and hence the same p-value, independently of all the other variants in the library. They will thus be classified the same way if the classification is based on the p-value. However, they will have different q-values (i.e. FDR), because the distributions of p-values differ between the two libraries, and would hence not be necessarily classified in the same way. By using a p-value threshold, we thus ensure that variants with similar mean effect and standard error are classified the same way. We use the FDRs associated to that p-value threshold (these FDRs are different in the two libraries) to subtract, in each library, the number of expected false discoveries from the total number of discoveries at that p-value threshold and hence report the number of true discoveries.

Reviewer #2:[…] A few comments follow. I am mostly concerned that better attention to analysis of error and reproducibility is necessary. I expect that it should be possible to address my comments with a handful of additional statistical analyses and more precise description in the text.1) Stochastic error in inferring growth rate from frequencies has not been adequate attention. The authors use all sequences with >10 input reads and >0 post-selection reads. Estimates of PPI scores (and, in turn, of epistasis) for genotypes with low read numbers will have considerable stochastic error associated with them, particularly if that error is propagated into products or ratios (as they are for epistatic effects). Although reproducibility may be relatively high overall, the authors should incorporate uncertainty caused by stochastic error into their classifications and quantifications of mutational and epistatic effects.

We agree and are well aware of this issue. We note however that since single mutants have a much higher coverage than double mutants (~20x more, see Materials and methods), the error on epistasis scores is mostly from the double mutant, had we used an error model, despite the product propagation of single mutants errors.

However, previous studies have been able to make new and significant discoveries without the use of such models. The fact that we find non-random enrichments for structurally related pairs of position shows that the increased noise added by those epistatic interactions derived from variants with low read numbers does not impede our ability to detect general trends.

To ensure that these results are not biased by these stochastic effects, we now show that they hold even when using only variants with a larger input read count (>100 in all three replicates; Figure 4—figure supplement 3C and Figure 5—figure supplement 5), which should decrease the stochasticity associated with low read counts.

2) The authors claim that reproducibility is high, and refer to a high rank-order correlation coefficient among replicates. The paper's major inferences, however, are not based on rank-order but on PPI score and statistics derived from it. What matters, then, is the reproducibility of the PPI score. The authors should instead report R^2^ for PPI among reps and between DMS and small-scale experiment. If the R^2^ is low, discussion of the implications for the authors' claims will be required.

We understand the reviewer’s concern. We initially used Spearman’s rank correlation because the distributions of PPI scores that are compared are not normal. We however now report Pearson’s correlation coefficient R as suggested. In all cases, the differences are negligible.

3) Because the library was created using doped degenerate codons rather than targeted synthesis, there are a great variety of genotypes in the library with different numbers of mutations. Paired-end sequencing is used (rather than bar-coding) to identify genotypes. Sequencing error may lead to mis-identification of genotypes, and this error may be biased towards genotypes with certain numbers of mutations. This concern does not appear to have been addressed. Reporting the degree of sequencing error and showing that it does not strongly affect the claims is necessary.

We agree and now report in the Materials and methods section “Sequencing data analysis and filtering” an estimate of the degree of sequencing errors based on the constant regions upstream of each leucine zipper (where the primer anneals to amplify the library) in the *trans* library:

“To estimate sequencing errors, we measured the per base error probability in the constant region upstream of the variable regions of FOS and JUN that were used as primer annealing sites for library amplification. […] Sequencing errors do not seem to affect the downstream estimate of PPI scores as filtering reads at a Phred score of 30, 35 or additionally filtering out reads where at least one of the called mutations had a Phred score below 30 (>90% of the wrongly called mutations in the constant region had a Phred score below 30 compared to < 25% of the correctly called bases) led to PPI scores that were highly correlated to the ones calculated using the above filters (Figure 1—figure supplement 1A).”

We also added Figure 1—figure supplement 1A that shows that filtering reads at a higher average Phred score or additionally filtering reads where the called mutations had a Phred score below 30 does not strongly affect the PPI scores (Pearson correlation of ~0.99).

For the *cis* library, the paired end reads overlap along the whole variable region making sequencing errors much less likely than in the *trans* library. However, this error cannot be accurately estimated as the paired end reads do not overlap over the constant region. We assume sequencing errors will not affect PPI scores in the *cis* library since the higher rate in the *trans* library does not.

4) I find the PPI score unintuitive. In theory, it might range asymmetrically around the value of 1 (wildtype) from 0 for a totally deleterious genotype to infinity. This alone bothers me (and could be addressed with a log-scale relative to wildtype). The assay itself puts practical limits on how much a mutant genotype might possibly increase the PPI, because there must be saturation point at which occupancy of the complex, given the dose of methotrexate in the assay, no longer improves fitness. The PPI numbers are therefore somewhat hard to interpret. I would prefer a transform that expresses the PPI on a log scale relative to wild type, rescaled so the intervals between the minimum and wild-type and that between wild-type and maximum are the same. I don't view the authors' method as fatally flawed, and I have no reason to think the conclusions would change if my method were used. I'd like the authors to consider the possibility that such an approach might make the numbers in the paper easier for the reader to interpret.

We understand the reviewer’s concerns and actually initially tried different ways of scoring variants, including rescaling. However, we believe that the present score is the most intuitive as it corresponds to the number of doublings each variant underwent and so relates to the selection coefficient (the selection coefficient would be equal to PPI score minus 1). This measure is widely used. Moreover, we believe, as does reviewer 1, that our score makes it intuitive to relate the growth rate in our assay to the thermodynamic model. Rescaling the data so that the difference between the wild-type and the minimum is the same as the difference between the wild-type and the maximum could potentially bias the data because there is no reason to think that the absolute change in complex concentration should be the same for the minimum and maximum. The position of wild-type on the sigmoidal function between PPI score and ΔΔG supports this view. This also shows that in practice, the PPI score cannot reach infinity, not because of saturation but because it is not possible to have more than 100% complex bound. The y-axis is indeed equivalent to the fraction of the limiting partner that is bound.

Regarding the saturation effects, we added the following sentences in the Discussion, to suggest that these might have little effects: “A substantial fraction of genetic interaction could however not been explained by structural contacts. […] Moreover, Levy et al. have shown using protein-complementation assay that the growth is correlated to complementation complex concentration over a wide-range of concentrations (Levy et al., 2014).”

To summarize, we believe that our score is relevant and intuitive because it relates the population dynamics of the competition assay to the biochemical/thermodynamic basis of the PPI.

5) I like the authors' efforts to quantitatively distinguish specific structural epistatic interactions from general thermodynamic epistasis caused by the nonlinear relationship between the proteins' affinity for each other and occupancy of the complex. Although the model has a nice theoretical basis and accounts for a substantial amount of epistasis, there may be additional causes of nonspecific epistasis remaining in the system – for example, if there are nonlinearities in the relationship between occupancy of the jun-fos complex and growth rate. I would therefore like to see the authors explore using Sailer and Harms' Genetics 2017 method, which identifies general epistasis in a way that is more agnostic to the specific biochemical causes or quantitative relationships, to determine if this is the case. If general epistasis can be identified using Sailer's approach even after the thermodynamic model is applied, this would be important, because it would further reduce (and probably sharpen) the cases of specific structural epistasis, and it would be an interesting finding, as well.

We agree with the reviewer and initially used a loess regression as an agnostic model instead of the thermodynamic model. The increase in percentage of variance explained is only of 0.4% on average across the three replicates of the whole *trans* library, 0.2% for the *trans* libraries restrained to single nucleotides changes and 0.5% for the *cis* library (and this using a pretty relaxed bandwidth parameter making the regression surface quite rugged, but however fitted by cross-validation). However, the thermodynamic model is both simpler and more interesting because it gives a biophysical explanation to the non-specific epistasis compared to an agnostic model. Also, since there were no major differences between the downstream results when using the two models, we decided to report only one model to save space. We chose the thermodynamic model for the reasons stated above.

6) I have some concern that the thermodynamic correction may affect cis and trans interactions differently. Stated a little too simply, the model is built around the expectation that mutations in trans that additively affect the energy of complex formation will have an exponential effect on occupancy of the complex and therefore on PPI; epistasis not accounted for by this relationship represent specific structural epistasis. It is not apparent to me that the same model appropriately corrects for general thermodynamic nonlinearity among cis- acting mutations. Suppose that two mutations within a protein independently affect the dG of the native structure; they will have an exponential effect on occupancy of the native structure. It is not obvious to me that the model will account for such nonlinear effects on the effective concentration of each molecule. If so, then pairs of cis-mutations that in fact act additively on the protein might be classified as structural specific interactions even after the thermodynamic model is applied. This could contribute to the apparent excess of classified epistasis in cis compared to in trans after removal of thermodynamic epistasis (Figure 5). If I am misinterpreting the model, it would be helpful for the authors to better explain in the main text what the model does and doesn't account for.

FOS and JUN, like most bZIP proteins, fold upon binding. This means that the whole reaction can be modeled as a two-state equilibrium between the unfolded proteins on one side and the complex on the other, with a single ΔΔG value for the whole process.

We added the following sentence: “Because leucine zippers, including Fos and Jun, fold upon binding (Patel et al., 1990, Thompson et al., 1993), the same thermodynamic model based on a two-state equilibrium between the two unfolded proteins and the complex can describe how mutations combine in *cis* as well as in *trans*.”

7) The authors classify the magnitude of PPI effects as strongly deleterious, weakly deleterious, neutral, and beneficial based on thresholds of 0.64, 0.96, and 1.04. The authors should provide a justification these numbers, which are presented as if they are arbitrary. (They do resemble some relevant numbers from the normal distribution, but the authors should make this explicit and justify their use.)

We agree that these thresholds are arbitrary. However we show in Figure 2—figure supplement 1 and Figure 5—figure supplement 2 that our conclusions hold when modifying the thresholds within relevant ranges for the class they delineate based on the distribution of PPI scores.

8) The interface of jun and fos appears as if it might be isologous, such that a mutation in one partner would be expected to have precisely the same structural and functional effect as the same mutation in the other (although this prediction would be compromised by divergence between the paralogs over time). The authors might discuss this, particularly as an explanation for the correlation between the effects of a mutation in one partner and the same mutation in the other. Isology would explain correlation; divergence would explain imperfect correlation.

That is indeed the point we tried to make. We now make this point more explicit by adding the following sentence: “This correlation indicates that the structural determinants of mutation effects in *FOS* and *JUN* remain well conserved despite sequence divergence over long evolutionary timescales.”

We did not employ the word “isolog” because we found only 15 references in PubMed and hence thought it could be considered as jargon to the wide readership of *eLife*.

9) Some bZip proteins, including fos and jun family members, can homodimerize. In the current assay, such events would interfere with growth, assuming that homodimers would compete with the heterodimer for the relevant subunits. Mutations that specifically affect the affinity of homodimerization – including those at the interface – would thus affect PPI, but not through trans-interactions. The authors should consider this possibility and note how it might contribute to their observations.

We now add the following paragraph in the Discussion to state how homodimerization could mitigate some of the results: “A substantial fraction of genetic interaction could however not be explained by structural contacts. […] Elucidating the remaining mechanisms of genetic interactions will thus require further studies that take these effects into account.”

10) The authors state that combining two mutations that positively affect growth rate leads to positive epistasis and cite Figure 3C. I believe that should be apparent as pies colored mostly green in the upper-right quadrant of the array. I don't see that in the data at all. That quadrant looks mostly yellow, suggesting little epistasis. What I believe is apparent in Figure 3C is that positive epistasis is concentrated in the rows where a weak beneficial mutation plus a deleterious mutation yield an outcome that is less deleterious than expected; this effect goes away when the thermodynamic model is applied (compare 3F). This is worth noting and, I hope, providing an explanation for.

We agree with the reviewer that two mutations that increase strength do not exhibit epistasis and deleted this statement. However, regarding the combination of strength-increasing and –decreasing mutations, we already state that “Positive genetic interactions are, however, generally detected between two mutations that greatly weaken the interaction and also often when combining strength-increasing and strength-decreasing mutations (Figure 3C).”

We now added the following sentence to explain how this effect is the result of the sigmoidal relationship between PPI score and ΔΔG and thus regressed out by the thermodynamic model: “Indeed, because of the sigmoidal nature of the model, a single mutant that decreases ΔΔG will increase PPI scores to a lower extent in the wild-type context rather than when combined with a mutation that destabilized the complex because of the saturation effect caused by the plateau of the sigmoid.”

Reviewer #3:[…] * The only thing that I see lacking experimentally is deep sequencing of the wildtype sequences to estimate the error rate. Right now there is no such estimate of how many of the observed mutations are actually due to library preparation of sequencing errors. This could easily be determined by sequencing wild type.

We agree that an estimate of sequencing error was lacking and now provide in the Materials and methods section “Sequencing data analysis and filtering” an estimate of the degree of sequencing errors based on the constant regions upstream of each leucine zipper (where the primer anneals to amplify the library) in the *trans* library:

“To estimate sequencing errors, we measured the per base error probability in the constant region upstream of the variable regions of FOS and JUN that were used as primer annealing sites for library amplification. […] Sequencing errors do not seem to affect the downstream estimate of PPI scores as filtering reads at a Phred score of 30, 35 or additionally filtering out reads where at least one of the called mutations had a Phred score below 30 (>90% of the wrongly called mutations in the constant region had a Phred score below 30 compared to < 25% of the correctly called bases) led to PPI scores that were highly correlated to the ones calculated using the above filters (Figure 1—figure supplement 1A).”

We also added Figure 1—figure supplement 1A that shows that filtering reads at a higher average Phred score or additionally filtering reads where the called mutations had a Phred score below 30 does not strongly affect the PPI scores (Pearson correlation of ~0.99).

For the *cis* library, the paired end reads overlap along the whole variable region making sequencing errors much less likely than in the trans library. However, this error cannot be accurately estimated as the paired end reads do not overlap over the constant region. We thus assume sequencing errors will not affect PPI scores in the *cis* library since the higher rate in the *trans* library does not.

* The computer code used for the data analysis should be included as a supplementary file and/or a GitHub repository (eLife now allows this).

We now uploaded the code on GitHub at https://github.com/gdiss/Diss_et_al_*eLife*_2018 and added the link at the beginning of the Materials and methods section.

* In various places (first in the subsection “Determinants of single mutant outcome”), the authors refer to the "identical substitutions" in both proteins. But they never clearly explain what this means. I'm assuming that the proteins are homologous, and they mean the homologous positions? This needs to be described much better.

We added the following sentence: “This correlation indicates that the structural determinants of mutation effects in *FOS* and *JUN* remain well conserved despite sequence divergence over long evolutionary timescales.”

* Unless I am misunderstanding, the thermodynamic model is in terms of mutational effects on the protein-protein binding. This makes sense for analyzing the trans mutational effects. But for analyzing the cis mutational effects, wouldn't we expect to instead be concerned with ddG values for the stabilities of the individual proteins?

FOS and JUN, like most bZIP, fold upon binding. This means that the whole reaction can be modeled as a two-state equilibrium between the unfolded proteins on one side and the complex on the other, with a single ΔΔG value for the whole process.

We added the following sentence: “Because leucine zippers, including Fos and Jun, fold upon binding (Patel et al., 1990, Thompson et al., 1993), the same thermodynamic model based on a two-state equilibrium between the two unfolded proteins and the complex can describe how mutations combine in cis as well as in trans.”